# Photochromic luminescence of organic crystals arising from subtle molecular rearrangement

Zihao Zhao [1], Yusong Cai[1], Qiang Zhang[1], Anze Li[1], Tianwen Zhu [1], Xiaohong Chen[1] & Wang Zhang Yuan [1] ✉

Photoluminescence (PL) colour-changing materials in response to photo-stimulus play an increasingly significant role in intelligent applications for their programmability. Nevertheless, current research mainly focuses on photochemical processes, with less attention to PL transformation through uniform aggregation mode adjustment. Here we show photochromic luminescence in organic crystals (e.g. dimethyl terephthalate) with PL varying from dark blue to purple, then to bright orange-red, and finally to red. This change is attributed to the emergence of clusters with red emission, which is barely achieved in single-benzene-based structures, thanks to the subtle molecular rearrangements prompted by light. Crucial to this process are the through-space electron interactions among molecules and moderate short contacts between ester groups. The irradiated crystals exhibit reversible PL transformation upon sufficient relaxation, showing promising applications in information storage and smart optoelectronic devices. This research contributes to the development of smart photochromic luminescent materials with significant PL colour transformations through molecular rearrangement.

Stimuli-responsive behaviour is one of the most fascinating characteristics of smart materials, which endows them with programmability, and thus showing prospective potentials in information storage and encryption, therapeutics as well as biotechnology[1–6]. Among various external stimuli such as solvent[7], mechanical stress[8], temperature[9] and pressure[10], light can be conveniently and rapidly controlled to actuate molecules with superior spatiotemporal accuracy[11–14]. Thus far, lots of photoresponsive materials together with respective mechanisms have been reported, including molecular photoswitches (e.g. azobenzene, spiropyran and diarylethene)[15–18], photoactivated phosphorescence via $^3O_2$ consumption[19–22] and photoinduced radical materials[23–26]. Nevertheless, most of them, especially those undergone photochemical reactions, experience colour change in appearance during photoluminescence (PL) transformation, which limits their applications in information encryption. Namely, once input, the information can be directly read without decoding

(e.g. photoexcitation). Furthermore, they typically have limited changes in PL colour. Additionally, their synthetic procedures are generally troublesome. Therefore, the development of intelligent and easily accessible photoresponsive materials with a wide range of colours still poses a significant challenge.

It has been widely acknowledged that molecular packing plays an essential role in organic solid-state emission[27], for which slight changes in molecular arrangement can lead to gigantic variations in photophysical properties[28]. For instance, different polymorphs of a compound often exhibit completely different emission behaviours[29,30]. Therefore, controlling the aggregation state offers a promising approach to achieve diverse luminescence. However, current methods for obtaining distinct aggregation states are primarily limited to artificial means, such as single crystal cultivation using various solvents. The spontaneous transformation between target aggregation states is rarely achieved[31–33]. Consequently, there is a pressing need to develop

[1]School of Chemistry and Chemical Engineering, Frontiers Science Center for Transformative Molecules, Shanghai Jiao Tong University, Shanghai, China.
✉e-mail: wzhyuan@sjtu.edu.cn

materials that can adjust their aggregation state in response to external stimuli, particularly light. This advancement would surely contribute to the development of intelligent luminescent materials.

Here, we report a surprising discovery of photoinduced PL transformation from crystalline dimethyl terephthalate (DMTPA) (Fig. 1a), which has been identified as a compound with crystallisation-induced phosphorescence and clustering-triggered emission (CTE) features[34,35]. It is newly found that DMTPA crystals undergo an intriguing photoinduced PL colour variation from dark blue to purple, then to bright orange-red, and finally to red upon UV exposure. Furthermore, after heat treatment or solvent fuming, the crystal can readily restore to its original blue emission (Fig. 1b). Remarkably, while the PL colour switching process goes immediately, the orange emission state can maintain for over 3 months under ambient conditions, demonstrating good stability. Further investigation reveals that subtle molecular rearrangement in crystals is responsible for the PL colour variation, which results in a metastable aggregation state with a lower energy gap evolving from the thermodynamically stable state. This photochromic luminescence (PCL)[36] is driven by UV irradiation (Fig. 1d) and can be reversed upon sufficient relaxation. Building upon these findings, we designed and synthesised additional analogues to verify the abovementioned mechanism (Fig. 1a), leading to the development of a series of PCL materials with varying photophysical properties. For instance, by replacing one ester group with carboxyl capable of forming strong hydrogen bonds, monomethyl terephthalate (MMTPA) crystal exhibits PCL in both prompt emission and moreover persistent room temperature phosphorescence (p-RTP; Fig. 1c). Compared to the previous PL colour-changing materials

prompted by intramolecular conformational adjustment[31,33,37], like the change of dihedral angle in benzil group with mechanical stumilus[33], the remarkable PCL of DMTPA crystals stems from conformational adjustment of aggregates, bringing about emerging emissive clusters without noticeable change at individual molecular level. Furthermore, in sharp contrast to the previous PCL systems, which are generally confined to the PL enhancement of already existing species[28,38], herein, distinct PCL is tightly associated with the dynamic evolution of emissive species. The discovery of such reversible photochromic luminescent systems through molecular rearrangement will contribute to the facile construction of smart luminescent materials for practical advanced technical applications.

## Results

### Photochromic luminescent behaviours of DMTPA crystals

DMTPA crystals demonstrate astonishing PL colour transformation under 312 nm UV irradiation, which turns from dark blue to purple in 10 s and continuously to bright orange in ~5 min (Fig. 2a and Supplementary Movie 1), while the crystal appearance remains unchanged. Notably, irradiation by other UV lights (e.g. 330 nm) can also trigger this PCL process. In situ emission spectra in Fig. 2b illustrate the dynamic change in PL of DMTPA crystals. Prior to UV exposure, the spectrum shows a single peak at ~420 nm, corresponding to blue emission with CIE (Commission Internationale de l'Eclairage) coordinate of (0.17, 0.09) (Fig. 2c). After 3 s of irradiation, three new peaks appear and gradually intensified at 595, 650 and 725 nm. Additionally, two inconspicuous peaks at 515 and 550 nm show slight increments, while the original peak (420 nm) significantly descends.

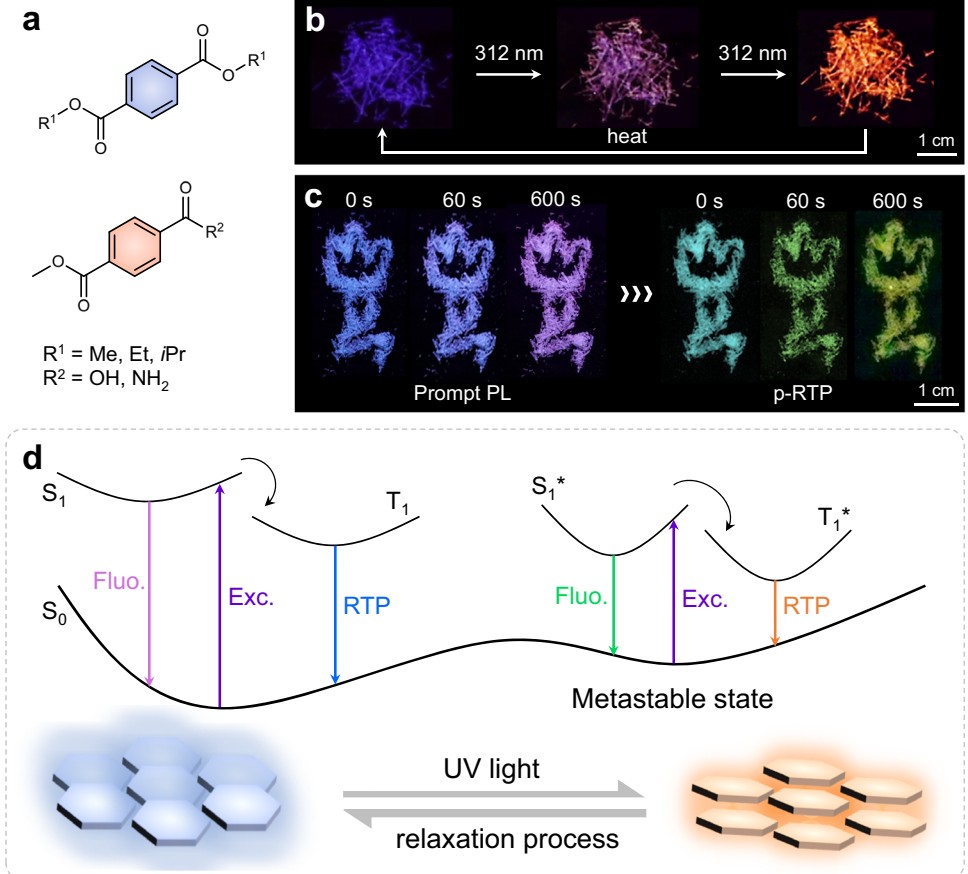

**Fig. 1 | Photochromic luminescent crystals with distinct PL colour change and proposed mechanism. a** Chemical structures of compounds studied in this work. **b, c** Luminescent photographs of colour transforming process of DMTPA (**b**) and MMTPA (**c**) crystals under 312 nm UV light. **d** Schematic illustration of proposed mechanism. UV light prompts subtle molecular rearrangement in crystals to a metastable state, which corresponds to lower energy gaps. Fluo., Phos. and Exc. represent fluorescence, phosphorescence and excitation, respectively.

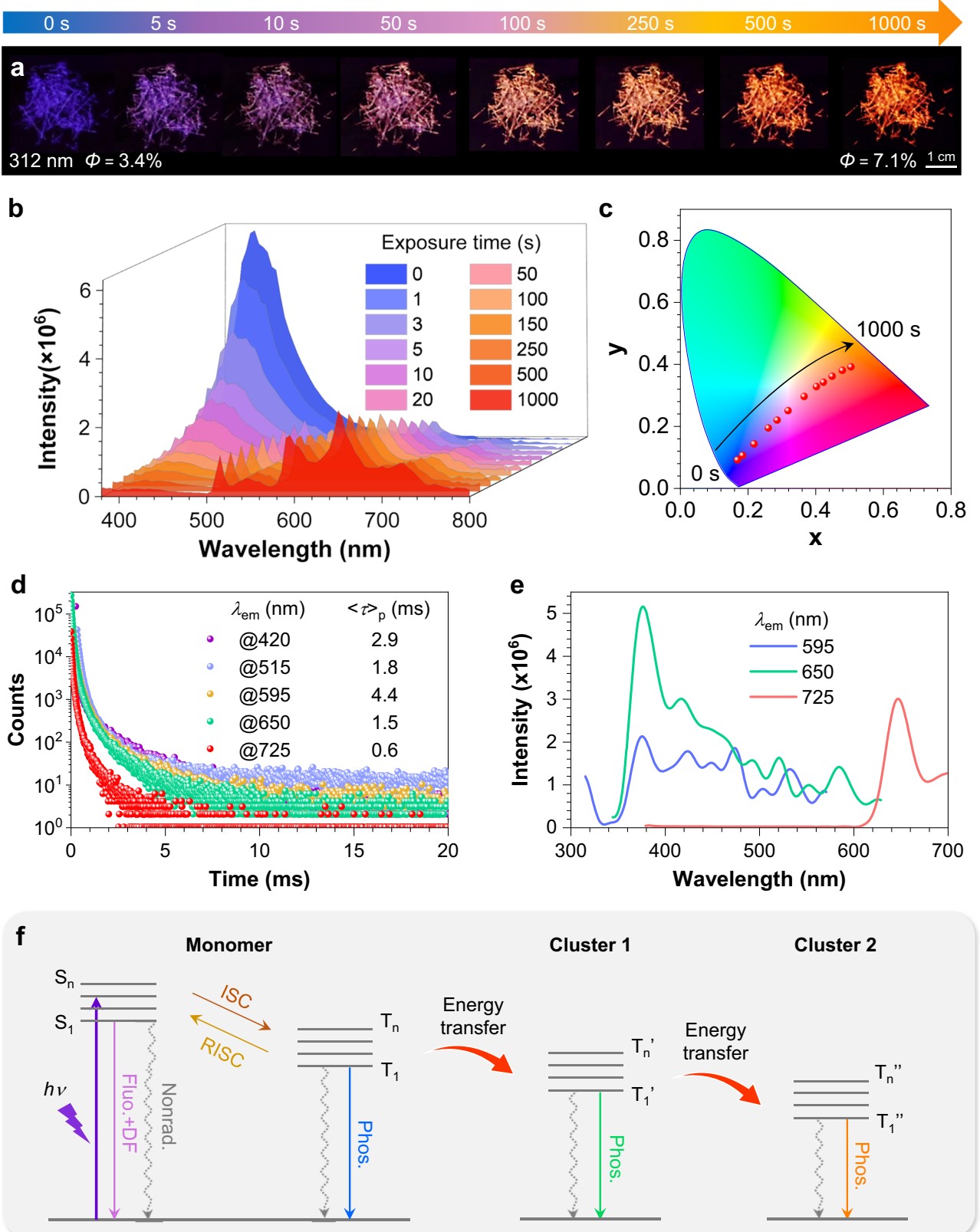

**Fig. 2 | Photophysical properties of DMTPA crystals. a–c** Luminescent photographs (**a**), prompt PL spectra (**b**) and CIE coordinate diagram (**c**) during the PCL process of DMTPA crystals under 312 nm UV irradiation with different exposure time. $\Phi$ is the quantum efficiency of total emission of the sample. **d** ms-Scale lifetimes for DMTPA crystals after 1000 s of 312 nm UV irradiation. **e** Excitation spectra of DMTPA crystals with varied emission wavelengths. **f** Jablonski diagrams illustrating the tandem energy transfer process. ISC, RISC, Fluo., Phos., DF and Nonrad. represent intersystem crossing, reverse intersystem crossing, fluorescence, phosphorescence, delayed fluorescence and nonradiative transitions, respectively. Source data are provided as a Source Data file.

Correspondingly, the PL colour shifts towards the purple region with CIE coordinate of (0.22, 0.13). This transformation process continues with prolonged irradiation, resulting in vibrant orange-red emission at ~1000 s with CIE coordinate of (0.50, 0.39). At this stage, the peak at ~420 nm gets negligible while the much redder emerging peaks become predominant. As a result, the PL quantum yield is enhanced from 3.4% to 7.1% (Fig. 2a). Notably, DMTPA crystals can retain this emitting property for over 3 months under ambient conditions, implying good stability of the newly generated emissive species (Supplementary Fig. 15).

Further UV exposure for 1 h gives continuously evolved PL colour, albeit at a much slower rate and with a relatively limited scope. Tangerine and deep red colours are noticed with successive irradiation of 5 and 12 h, respectively. Subsequently, the PL colour remains almost unchanged, indicating a balance of the PCL process (Supplementary Fig. 16). Remarkably, the two peaks at 515 and 550 nm, generated during the early PCL stage, gradually diminish, providing further evidence for the dynamic evolvement of emissive species. For the lack of hydrogen donors, possible photochemical reactions like hydrolyzation or photolysis engendering cyclohexadienyl radicals are excluded[39], as confirmed by the invariant $^1$H and $^{13}$C NMR and HPLC measurements before and after UV irradiation (Supplementary Figs. 1, 2 and 17 and Supplementary Note 1). Moreover, no photoinduced radical arising from bond breaking is observed, as indicated by the unchanged electron paramagnetic resonance (EPR) spectra (Supplementary Fig. 18). The consumption of $^3O_2$ is also ruled out, since the same PCL phenomenon can be observed in a vacuum (Supplementary Fig. 19). Therefore, it can be inferred that certain physical changes are responsible for the PL transformation.

Significantly, the delayed PL spectra of DMTPA crystals are basically consistent with the prompt ones (Supplementary Fig. 20). Notably, millisecond (ms)-scale lifetimes for nearly all peaks ranging from 420 to 725 nm are detected (Fig. 2d and Supplementary Fig. 21a), nanosecond (ns)-scale lifetimes, however, are merely detected at ~420 and 515 nm, indicative of fluorescence-phosphorescence dual emission at ~420 and 515 nm, and pure phosphorescence for others (Supplementary Fig. 21b, c). Furthermore, prompt and delayed PL spectra of $10^{-5}$ M DMTPA/2-MTHF solution at 77 K reveal the fluorescence and phosphorescence for monomeric DMTPA occurring at 302 and 420 nm, respectively (Supplementary Fig. 22). It is also noted that DMTPA dimers fluoresce UV at around 374 nm and diverse clusters widely exist in its aggregates (Supplementary Fig. 23 and Supplementary Note 2)[35]. Therefore, it can be inferred that the broad peak at around 420 nm comprises monomeric phosphorescence and aggregate fluorescence of DMTPA, while the photogenerated ones at redder region are attributable to phosphorescence from clustered molecules[35]. Impressively, remarkable red or even NIR emission are readily achieved through the PCL process, which is rarely realised in single-benzene based compounds. Consequently, the photoinduced PL colour variation, namely PCL, is highly correlated with the emerging DMTPA clusters. This conclusion is further supported by the absorption of DMTPA crystals (Supplementary Fig. 24), in which a peak at 350 nm arises as the exposure time increases, implying newly generated aggregates with extended conjugation.

In addition to the emerging peaks upon UV exposure, there is a noticeable decline in PL intensity at ~420 nm (Fig. 2b). To address it, excitation spectra of DMTPA crystals were measured (Fig. 2e). While an intense peak at exactly 420 nm is observed with the emission wavelengths of 595 and 650 nm, the optimal excitation, for the 725 nm emission, is 650 nm. This result suggests a potential tandem energy transfer process (Fig. 2f). During the PCL process, the tandem energy transfer readily occurs from the monomer excitons (420 nm, energy donor) to the photogenerated clusters emitting at 595 and 650 nm (cluster 1, intermediate, both energy acceptor and donor) and then to the clusters 2 luminescing at 725 nm (energy acceptor). The shortened

ms-scale lifetime at 420 nm, after PL transformation, validates this energy transfer process (Fig. 2d and Supplementary Fig. 21a). It gains additional support from the emission spectrum with 420 nm light excitation, which can be deconvoluted into four peaks locating at 591, 632, 659 and 706 nm (Supplementary Fig. 25), basically consistent with those excited by 312 nm UV light. As aggregates accumulate in response to prolonged photostimulus, the consumption of emission at 420 nm will progressively increase, thus leading to the attenuation at ~420 nm.

## Understanding the mechanism of PCL

As discussed above, newly generated aggregates are responsible for the PCL phenomenon. PL intensities of photogenerated peaks (e.g., 650 and 725 nm) scales linearly with the power density of UV irradiation without saturation (Supplementary Fig. 26 and Supplementary Note 3), excluding the mechanism of photoinduced clustering in lattice defects[40–42]. Therefore, these new aggregates should derive from the conversion of the original, probably through the conformational adjustment at excited states[28,33,37,38]. To provide more compelling evidences, we conducted characterisations focusing on variations in intermolecular interactions.

Firstly, DMTPA crystals before and after photoirradiation were characterised by X-ray diffraction (XRD) technology (Fig. 3a). The exposure time under 312 nm UV light was extended to 12 h to accumulate the possible structural variations. While the diffraction peaks corresponding to (111), (112) and (002) faces disappeared, a new weak peak corresponding to (119) diffraction emerged, suggesting the adjustments in molecular alignment within crystals. Meanwhile, Raman spectroscopy is known for its sensitivity in detecting slight morphological variations of molecular conformations on account of its ultra-surface-sensitivity[43,44]. As demonstrated in Fig. 3b and Supplementary Fig. 27, the relative Raman intensity of the peak at 3085 cm$^{-1}$, which corresponds to the stretching vibration of aromatic C−H bonds, remains intact after 5 min of 312 nm UV irradiation, despite the PL transformation has simultaneously occurred. However, with continuous irradiation for 12 h, its relative intensity remarkably descends, indicating the change in intermolecular interactions, particularly π-π interactions. These results indicate that the molecular rearrangement is too minuscule to be perceived in the early PCL process, while is accumulated and become detectable with prolonged irradiation. Additionally, the peak at 87 cm$^{-1}$ is shifted to 96 cm$^{-1}$ upon continuous UV stimulus (Fig. 3b), implying increased restrictions on molecular motions[45]. Such restrictions effectively suppress nonradiative decay processes and stabilise triplet excitons, thus favoring for the RTP emission. Further in situ FTIR spectra, which focus on the skeletal vibrations of aromatic C=C bonds, disclose an upward trend in characteristic peaks at 1608, 1535 and 1503 cm$^{-1}$ with prolonged exposure time (Fig. 3c and Supplementary Fig. 27)[45]. This trend aligns with the Raman result and provides further confirmation of the enhanced π-π interactions upon UV irradiation.

To probe the structure change of DMTPA crystals, single-crystal XRD analysis was further performed (Fig. 3d and Supplementary Fig. 28a, b). While there is no face-to-face π-π stacking among molecules due to the excessive distance (Fig. 3d, top), abundant edge-to-face π-π interactions are found, thanks to the tight molecular packing of DMTPA (Fig. 3d, bottom)[46,47]. Such interactions are duly verified by the theoretical calculation of noncovalent interactions (NCI) analysis (Fig. 3d, Supplementary Fig. 28c and Supplementary Note 4). Despite being weaker than the face-to-face interactions, the extensive edge-to-face π-π interactions, together with other short contacts, establish an efficient through-space conjugation (TSC) network that significantly facilitates redder emissions. These intermolecular interactions and TSC prevent drastic variations in the crystal structure, while still allowing subtle changes in molecular alignment upon irradiation, as confirmed by the in situ single crystal analysis. The dihedral angle

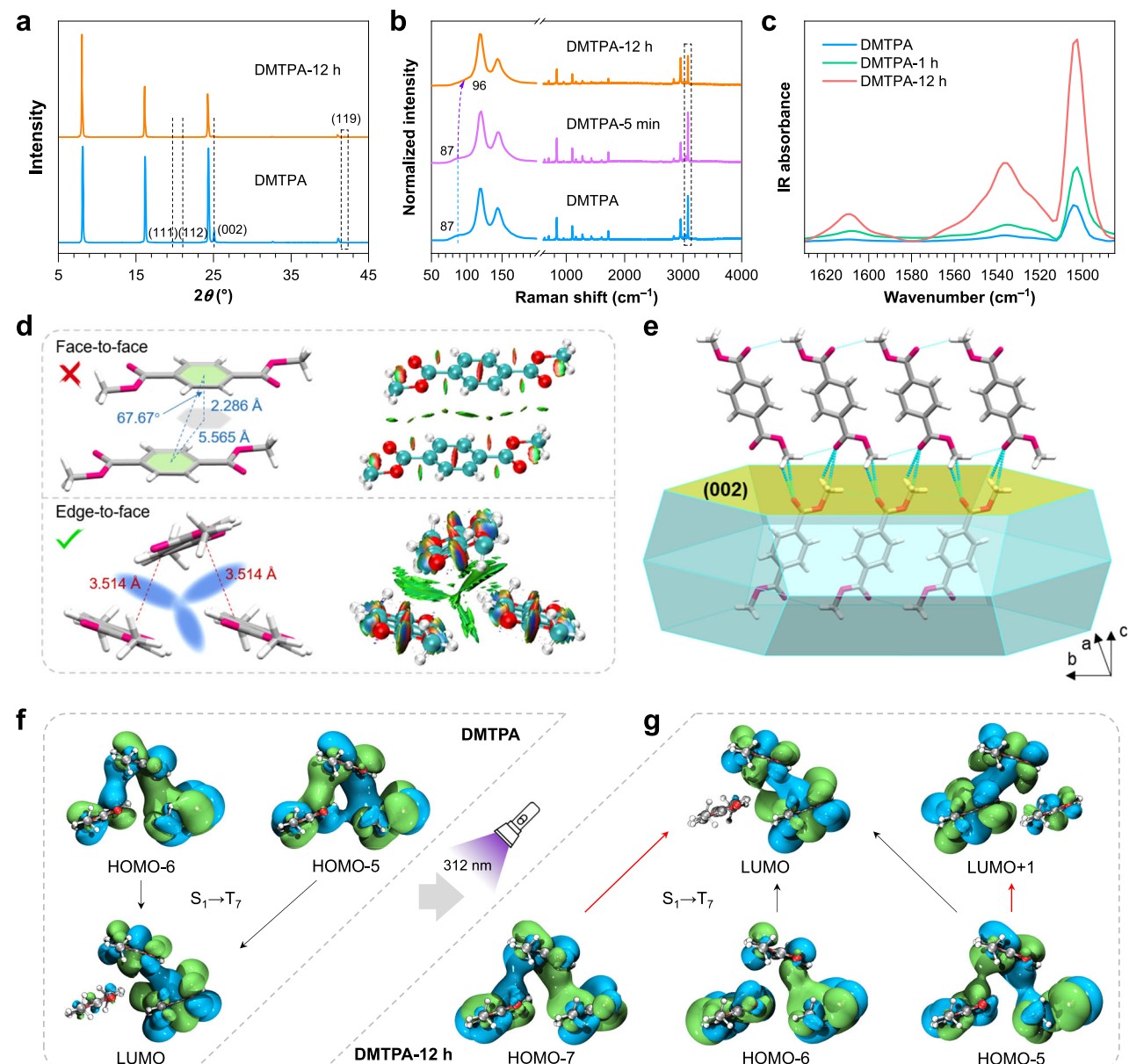

**Fig. 3 | Characterisations on molecular arrangement and interactions of DMTPA crystals. a–c** XRD patterns (**a**), Raman spectra (**b**) and FTIR spectra (**c**) of DMTPA crystals with different exposure time under 312 nm UV irradiation. **d** Single crystal structure and fragmental molecular packing of DMTPA as well as identification of π–π interactions. The red, green and blue surfaces represent strong repulsion, van der Waals interaction and strong attraction, respectively. **e** Predicted crystal morphology and intermolecular interactions corresponding to the (002) face. **f**, **g** Electron densities of the HOMO and LUMO levels for DMTPA trimer before (**f**) and after (**g**) 12 h of 312 nm UV irradiation. Source data are provided as a Source Data file.

between two DMTPA molecules involved in edge-to-face π-π interactions gradually decreases during the PCL process (Supplementary Fig. 29), and the bond lengths of C=O•••H−C short contacts between longitudinally arranged molecules shorten from 2.773 to 2.770 Å in 5 min, and then to 2.762 Å in 12 h (Supplementary Fig. 30). These observations, alongside the shortened distances (from 3.514 to 3.495 and then to 3.488 Å) of the edge-to-face π-π interactions (Supplementary Fig. 31), indicate an enhancement of π−π interactions, which agrees well with the FTIR results.

Notably, not all intermolecular interactions become stronger in response to photostimulus. The short contacts between transversely arranged DMTPA molecules, specifically C=O•••C and C=O•••H−C interactions, gradually lengthen from 3.183 and 2.815 Å to 3.185 and 2.820 Å, respectively (Supplementary Fig. 30). Importantly, despite the presence of conflict effects, during this transformation, the crystal unit

cell contracts, as evidenced by the reduction in the edge lengths while the density increases (Supplementary Table 1). These overall changes along with enhanced TSC account for the long-wavelength emission of the rearranged aggregates. Despite being subtle, these photoinduced rearrangements are transmitted to surrounding molecules within the dense and orderly periodic TSC network, resulting in synergistic amplification effect for the generation of significant PL transformation[48]. Furthermore, by locally exciting the DMTPA crystal, the PCL process propagates from the irradiated area to the unirradiated region (Supplementary Fig. 32 and Supplementary Note 5), duly supporting such amplification effect.

To further investigate varied intermolecular interactions, growth morphology modelling utilising the Bravais-Friedel Donnay-Harker (BFDH) method was performed based on 2θ scans and single crystal XRD data using the Mercury package[49]. The drastic decrease in

intensity observed at $2\theta$ of 25.1° indicates a variation in molecular arrangement on the (002) face (Fig. 3a), which corresponds to C=O•••C and C=O•••H−C short contacts that have been shown to change (Fig. 3e and Supplementary Fig. 30). The weakening of these contacts is also evident in the NCI analysis plot (Supplementary Fig. 33). Similarly, the depressed peak at $2\theta$ of 19.7° also implies the change of molecular arrangement along the (111) face, corresponding to the edge-to-face π-π interactions and C=O•••H−C contacts between longitudinally arrayed molecules (Supplementary Fig. 34). These results elucidate variations in multiple interactions around DMTPA molecules in response to photostimuli, providing additional evidence for photoresponsive molecular rearrangement.

Furthermore, TD-DFT theoretical calculations were performed on DMTPA trimers extracted from the single crystal structure. Before photostimulus, two intersystem crossing (ISC) channels from $S_1$ to $T_7$ are observed, with an energy gap of 0.308 eV (Supplementary Fig. 35). HOMOs show through-space edge-to-face π-π interactions distributed across the trimer, whereas the LUMO is mainly located at two molecules, indicative of a partial charge transfer feature (Fig. 3f). After 312 nm UV irradiation, the trimer exhibits four ISC channels, still from $S_1$ to $T_7$, with a decreased energy gap of 0.306 eV (Fig. 3g and Supplementary Fig. 36). Increased ISC channels and narrowed energy gaps ($\Delta E_{ST}$) are favourable for the phosphorescence, which is consistent with the experimental results.

Derived from the above discussions, the mechanism underlying PCL of DMTPA crystals can be concluded as follows: external photostimulus drives the subtle molecular rearrangement, which is amplified by the compact and orderly periodic TSC network throughout the DMTPA crystal lattice, thereby forming new clustered luminophores with distinct PL emissions. Additionally, moderate short contacts between transversely arranged molecules from ester groups are also necessary for empowering molecular reorganisation under mild stimuli. Moreover, the crystalline nature of DMTPA is also necessary for PCL, which is crucial for the formation of periodic TSC network and subsequent amplification effect. While the liquid nitrogen quenched DMTPA solid (also crystalline) shows evidently decreased PCL property, its ground powder (almost amorphous) demonstrates neglectable change upon UV irradiation (Supplementary Fig. 37 and Supplementary Note 6).

## Universality of PCL phenomenon and mechanism verification

To investigated the universality of the PCL phenomenon and validate its underlying mechanism, we examined diethyl terephthalate (DETPA) and diisopropyl terephthalate (DiPTPA) with larger alkyls, which are effective in tuning the π-π interactions. Despite both compounds demonstrate variations in PL spectra upon irradiation (Supplementary Figs. 3, 4, 38 and 39), only DETPA crystals show colour-changing PL, transitioning from blue to purple after 10 min of irradiation, whereas the emission of DiPTPA crystals gradually fades from bright light blue (Fig. 4a). Concretely, other than the decrease in intensity at ~440 nm, DETPA crystals show the emergence of three peaks at 605, 665 and 730 nm emerge in the prompt PL (Fig. 4b). In contrast, for DiPTPA crystals, only an attenuation at ~420 nm is observed (Fig. 4c). Correspondingly, after 10 min of irradiation, DETPA crystals demonstrate a slightly increased quantum yield from 2.9% to 3.2%, while that of DiPTPA drastically decreases from 12.6% to 2.1% (Fig. 4a). Moreover, ms-scale lifetime measurement reveals the predominant triplet feature for all peaks (Supplementary Figs. 40–43), with the emerging ones exhibiting no ns-scale components (Supplementary Fig. 44), illustrating their RTP nature. Furthermore, variations in XRD patterns confirm the occurrence of molecular rearrangement in both crystals (Supplementary Fig. 45), which is further verified by the in situ single-crystal analysis (Supplementary Figs. 46 and 47 and Supplementary Tables 2 and 3).

However, although both DETPA and DiPTPA crystals are proved to undergo molecular rearrangement by XRD analysis, the through-space electron interactions in the latter are not enough to provide an amplification effect due to the strong steric hindrance of isopropyls. As demonstrated in Fig. 4f, widespread face-to-face π−π interactions are observed in DETPA crystals, whereas there is no such contact among DiPTPA molecules. Furthermore, the characteristic IR peaks of DETPA crystals associating skeletal vibration of aromatics rise evidently after 12 h of UV stimulus (Fig. 4d and Supplementary Fig. 48), illustrating the photoinduced enhancement of π-π interactions. In contrast, those of DiPTPA crystals depicts tiny changes (Fig. 4e and Supplementary Fig. 48), suggesting inappreciable variation of π-π interactions. In this sense, the amplification effect deriving from the molecular aggregation and through-space electron interactions play an indispensable role in generating new emissive species, without which subtle molecular rearrangements are insufficient to prompt the PCL process.

Nevertheless, excessive intermolecular interactions can prevent molecular rearrangement. For example, 2,6-dimethyl 2,6-naphthalenedicarboxylate (DMNDCA), despite owning a similar crystal structure to DMTPA (Supplementary Fig. 49), only demonstrates unremarkable PL colour transformation from dark purple to violet even after 20 min of UV exposure, accompanying negligible newly generated PL peaks (Supplementary Fig. 50). Subtle molecular rearrangement can be noticed from DMNDCA after 12 h of accumulation (Supplementary Figs. 51 and 52 and Supplementary Table 4), the process, however, is significantly more difficult compared to that observed in DMTPA crystals.

It should be stressed that the amplification effect empowered by through-space electron interactions is not the only prerequisite of PCL. Sufficient molecular motions are also crucial, as suggested by the absence of PCL phenomenon of DMTPA crystals at 77 K (Supplementary Fig. 53). To explore this further, MMTPA and terephthalic acid (TPA) crystals are chosen for comparison, for which strong hydrogen bonds are expected. Impressively, upon 10 min of 312 nm UV irradiation, MMTPA crystals not only exhibit PL colour changing from dark blue to bright magenta, but also display tunable p-RTP afterglow varying from cyan to green and then to yellow (Fig. 5a, b and Supplementary Fig. 54), thanks to the hydrogen bonding stabilised triplet excitons. To the best of our knowledge, this is the first example showing p-RTP afterglow with remarkable PCL through the molecular rearrangement mechanism. Correspondingly, the peaks at ~430 and ~530 nm in the prompt PL spectra descend with prolonged exposure time, while those being simultaneously increased at 600, 655 and 720 nm (Fig. 5b), accompanying CIE coordinate shifting from (0.20, 0.19) to (0.37, 0.24) (Fig. 5c) and quantum yield increasing from 5.4% to 9.1% (Fig. 5a).

Distinct from DMTPA, the peak at ~530 nm in the delayed PL spectra remain predominate after UV irradiation (Fig. 5d), thus leading to p-RTP located in the upper right region of the CIE diagram (Fig. 5c). Meanwhile, both prompt and delayed emissions at 600, 655 and 720 nm also contribute relatively small fractions to the whole PL, suggesting that stronger short contacts somehow inhibit the PL transformation process. The photoinduced molecular rearrangement mechanism is still applicable to the PCL of MMTPA crystals and can be verified through similar characterisations, including in situ FTIR spectra (Fig. 5e and Supplementary Fig. 55), XRD patterns (Supplementary Fig. 56) as well as the single-crystal analysis (Supplementary Fig. 57 and Supplementary Table 5). Therefore, by introducing stronger hydrogen bonds, MMTPA crystals can also demonstrate PCL phenomenon and moreover with distinct p-RTP colour transformations, thus shedding lights on the fabrication of novel multicolour p-RTP smart materials with PCL property. In contrast, TPA crystals show hardly any change in PL after 10 min of 312 nm UV irradiation (Fig. 5a and Supplementary Fig. 58), along with virtually unaltered quantum

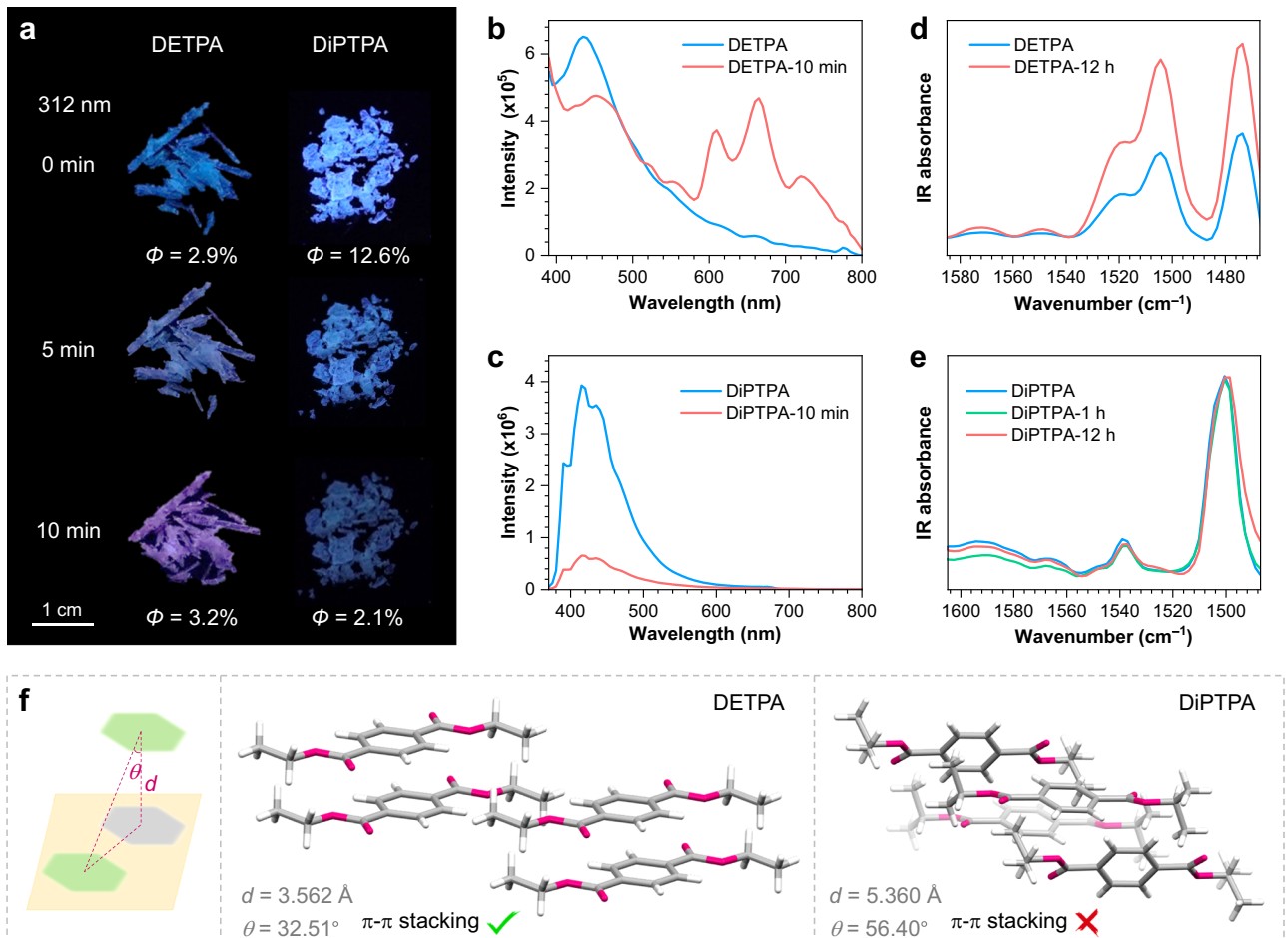

**Fig. 4 | Photoresponsive property of DETPA and DiPTPA crystals.**
**a**−**c** Luminescent photographs (**a**) and prompt emission spectra (**b**, **c**) of PL transformation process of DETPA and DiPTPA crystals under 312 nm UV irradiation with different exposure time. *Φ* is the quantum efficiency of total emission of the sample. **d**, **e** FTIR spectra of DETPA (**d**) and DiPTPA (**e**) crystals with different exposure time under 312 nm UV irradiation. **f** Single crystal structure and fragmental molecular packing of DETPA and DiPTPA. Source data are provided as a Source Data file.

yield (Fig. 5a), XRD patterns (Supplementary Fig. 59) and FTIR spectra (Supplementary Fig. 60).

Clearly, as ester groups are gradually replaced by carboxyl units, the crystals tend to generate much longer and more efficient p-RTP (Supplementary Fig. 61) with attenuated PCL behaviours. To uncover the reason behind this evolution, their single-crystal structures were compared (Fig. 5f). In DMTPA crystals, molecules are loosely packed due to the weak short contacts between ester groups, making them more prone to rearrange in response of external photostimulus, thus readily to proceed PCL. However, such weak intermolecular interactions cannot fully restrain nonradiative transitions, thus leading to relatively short RTP. For MMTPA, two molecules form an integral dimer through multiple hydrogen bonds between adjacent carboxyls (Supplementary Fig. 62). Thanks to the ester groups at each end of the dimers, they can arrange in a manner similar to DMTPA molecules, thus retaining the PCL ability. Furthermore, the hydrogen bonds can effectively stabilise the triplets, thereby leading to photochromic p-RTP. In contrast, TPA molecules are tightly connected via diverse short contacts including multiple hydrogen bonds, π−π interactions and others, forming a compactly packed planar structure with ample interlayer interactions (Supplementary Fig. 63)[50], which effectively prohibit molecular rearrangement upon photoirradiation while promoting p-RTP emission. These results strongly imply that moderate short contacts between transversely arranged molecules is a key factor empowering molecular reorganisation in response to photostimulus. This notion is additionally supported by the presence and absence of

PCL in methyl 4-(aminocarbonyl)benzoate and terephthalamide crystals (Supplementary Fig. 64), respectively.

As indicated above, the amplification effect facilitated by through-space electron interactions plays a crucial role in PCL process, which enables even subtle molecular rearrangements to trigger the formation of newly clustered emissive species. Additionally, moderate short contacts are necessary for allowing such molecular reorganisation in response to mild photostimulus. Inspired by this mechanism, it is feasible to regulate the colour range and response time of PCL by appropriately adjusting the moieties. Moreover, the integration of p-RTP with PCL becomes achievable by introducing stronger short contacts. These approaches ensure the rational design of more functional photoresponsive PL materials.

## Reversibility and applications

As exhibited in Fig. 6a, photoresponsive DMTPA crystals can resume blue emission after heating for 10 min at 80 °C in air, and the PL transformation process towards orange emission can be repeated with further 312 nm UV irradiation, illustrating excellent reversibility. On the other hand, since the PCL property of these compounds are highly dependent on the crystal structure (Supplementary Fig. 37), grinding is another way to reawaken the blue emission from DMTPA crystals. Upon solvent fuming and consequent UV irradiation, the ground powders show orange PL again. Furthermore, the emission spectra during the PL colour variation-recovery processes disclose the almost identical profiles of the photoirradiated samples, while the

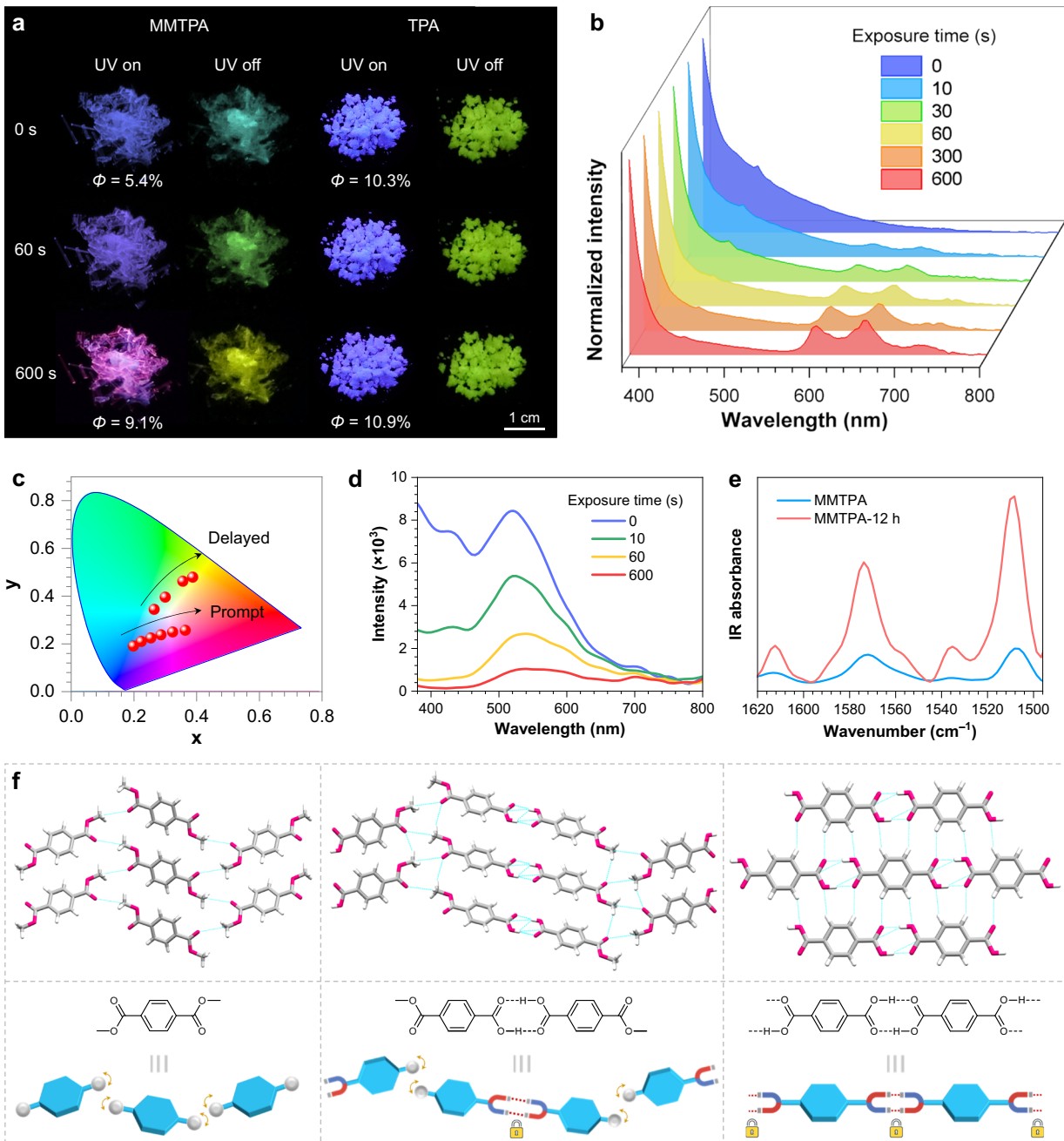

**Fig. 5 | Photoresponsive property of MMTPA and TPA crystals. a** Luminescent photographs of MMTPA and TPA crystals under and after ceasing the 312 nm UV irradiation with different exposure time. $\Phi$ is the quantum efficiency of total emission of the sample. **b**–**d** Prompt emission spectra (**b**), CIE coordinate diagram (**c**) and delayed ($t_d = 1$ ms) emission spectra (**d**) of PL colour transformation process of MMTPA crystals. **e** FTIR spectra of MMTPA crystals before and after 12 h of 312 nm UV irradiation. **f** Schematic fragmental molecular packing of TPA, MMTPA and DMTPA crystals. Source data are provided as a Source Data file.

main peaks of ground or heated crystals hypsochromically shifted with comparison to original DMTPA crystals (Fig. 6b). This blue-shift should be ascribed to the destruction of crystal structure that attenuates conformational rigidity, causing the decrement of monomeric phosphorescence at ~420 nm. Additionally, the orange PL of transformed DMTPA crystals can spontaneously revert to purple when placed at ambient conditions for over 3 months (Supplementary Fig. 15). This reversal process indicates that the molecular rearrangement from orange-emitting aggregates to blue-emitting ones is thermodynamically permissible and that the former is a metastable state generated in response to external UV stimulus (Fig. 1d).

The unique PCL behaviours of DMTPA crystals endow them promising applications in smart encryption ink and information storage. For instance, a 5×4 cm² butterfly pattern was printed on a filter paper through simple screen printing method with a mixture of commercially available aloe gel and DMTPA/EA (right wing) or DiPTPA/EA (left wing) solutions (Fig. 6c). The initial PL from both wings is similar. However, upon exposure to 312 nm UV light for 1 min, the PL from the right wing becomes purple, and further turns to conspicuous orange-red with increasing exposure time of 3 min (Fig. 6d). In contrast, the PL of the left wing gradually fades away. This experiment clearly demonstrates the feasibility of using DMTPA as intelligent encryption ink.

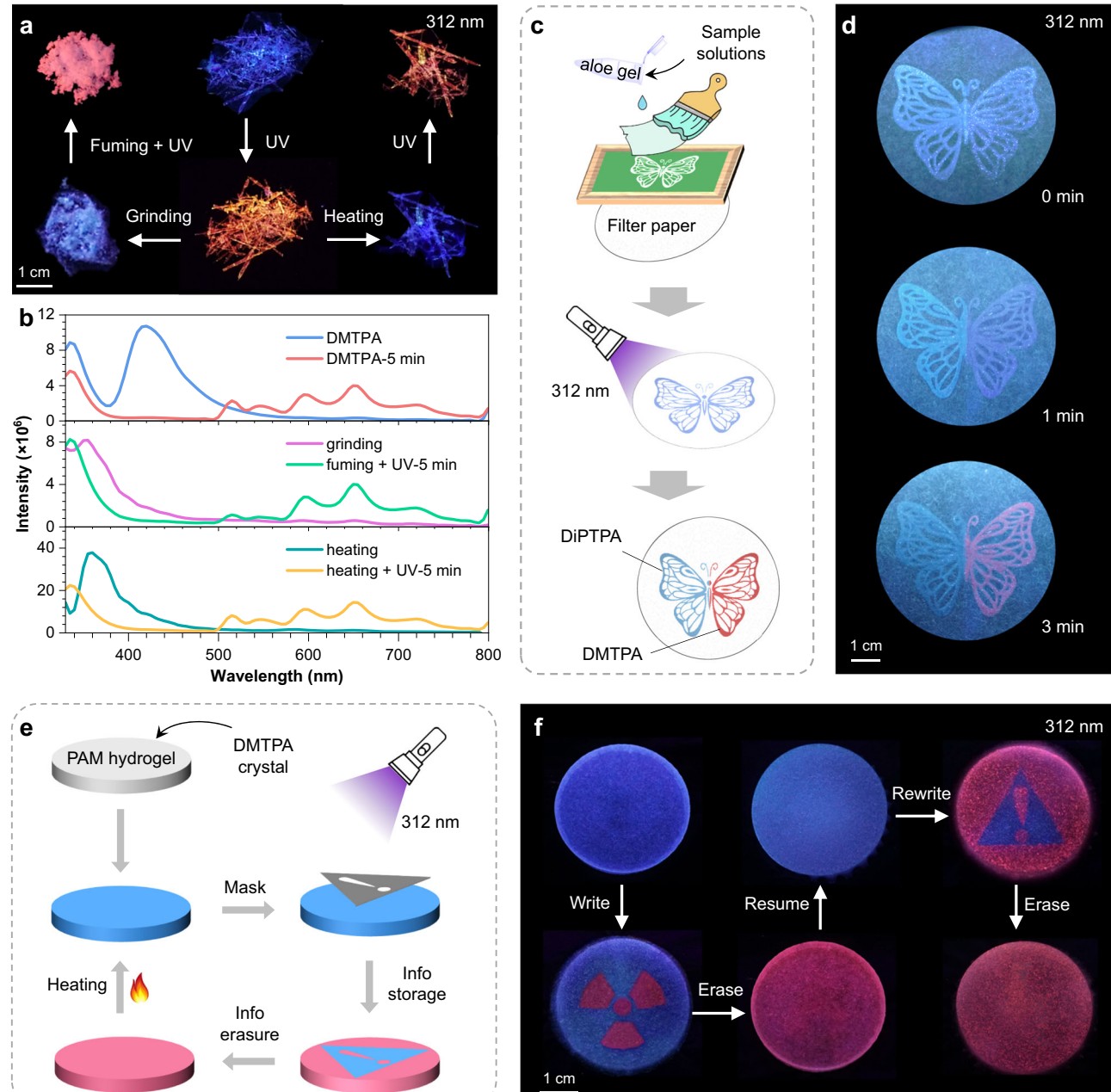

**Fig. 6 | Reversibility of PL transformation and applications demonstration.**
**a**, **b** Luminescent photographs (**a**) and prompt emission spectra (**b**) of DMTPA after different physical treatments. The orange-emitting DMTPA crystals after PCL process can resume blue emission after grinding or heating treatment, and repeat the PLC process upon (solvent fuming and) UV irradiation. **c** Schematic diagram of the procedure of screen printing and subsequent UV disposal. **d** Luminescent photographs of the printed pattern under 312 nm UV irradiation with different exposure time. **e** Schematic diagram of preparation, information input, erasure and recycle of DMTPA/PAM hydrogel. **f** Luminescent photographs of DMTPA/PAM hydrogel during the process of information input, erasure and recycle under 312 nm UV irradiation. Source data are provided as a Source Data file.

However, the crystalline nature and thus nondeformability and fragility of DMTPA crystals limited their applications. To overcome this limitation, we combined DMTPA crystals with polyacrylamide (PAM) hydrogel to create a flexible and reusable material featuring photo-responsive PL transformation for information storage and erasure. As schemed in Fig. 6e, tiny DMTPA crystals were dispersed into an aqueous PAM solution to prepare the hydrogel that emits blue PL under 312 nm UV light. By exposing specific areas through a mask for ~5 min followed by removal of the mask, red emission is observed in the exposed part while the rest of the hydrogel remains blue, thus enabling dynamic information storage. The input pattern can be easily erased by further UV irradiation to transform the entire hydrogel into a red-emitting state. After erasing the information, the hydrogel can be reused by heating it at 80 °C for 10 min to restore its blue PL. This PCL and restoring processes can be repeated for 10 cycles (Supplementary Fig. 65). The practical application of this hydrogel is shown in Fig. 6f, demonstrating its potential prospects in intelligent information carriers and flexible devices utilising the intriguing PCL phenomenon.

## Discussion

In summary, we have demonstrated the unique PCL phenomenon of DMTPA crystals from dark blue to purple then to bright orange and finally to deep red PL upon UV irradiation. Through various characterisations, we have uncovered that UV irradiation induces subtle

molecular rearrangement by examining the variations in interactions among DMTPA molecules. This photoinduced molecular rearrangement is amplified via the compact and orderly periodic through-space electron interactions (e.g. π–π interactions) within the crystal lattice, driving the original aggregates to transform into new clusters with much redder emissions. Furthermore, moderate short contacts between ester groups are also necessary for enabling DMTPA molecules to rearrange under mild external photostimulation. We further designed and synthesised additional derivatives to verify the significance of through-space electron interactions and moderate short contacts, thereby validating our proposed mechanism and enabling the development of more PCL materials with versatile functions (e.g. p-RTP). Moreover, such PCL process can be reversed through sufficient relaxations like grinding-fuming and heating treatments. Based on these findings, we developed a smart anti-counterfeiting ink for screen printing as well as a flexible colour-changing hydrogel. This research may spur further exploration of smart luminescent materials with PCL characteristics, thus paving the way for prospective applications towards intelligent optoelectronic devices.

## Methods

### Reagents and materials
All reagents for photoluminescence measurements were purified by recrystallisation and/or column chromatography before use to guarantee their purity. Unless specified, all characterisations were carried out under ambient conditions.

### Irradiation experiments
312 nm UV lamp (Spectroline EB-280C, Spectronics Corporation, USA) with two tubes (power: 8 watts each) is utilised as excitation light source during the photochromic luminescent process. The power density of 312 nm UV irradiation on the samples is ~3.5 mW/cm$^2$. It is worth noting that lower power densities are also capable of driving the PCL process, only with longer response time. The quantity of samples does not need to be fixed, which is determined by the required amounts for different characterisations. For photography, ~30 mg samples are used.

### Preparation of DMTPA supercooled liquid and its ground powders
DMTPA crystals were melted through heating to 140 °C, and the acquired liquid was quickly put into liquid nitrogen to obtain DMTPA supercooled liquid. The supercooled liquid (white solid) was ground into powders to further reduce its degree of crystallisation.

### Preparation of DMTPA/PAM hydrogel
The monomer of PAM, acrylamide, was purified before polymerisation. It was firstly dissolved in ethanol to form a concentrated solution, which was consequently added into n-hexane dropwise while vigorously stirring. The resulting precipitates were then collected by filtration and redissolved in ethanol. This dissolving-precipitation process was repeated three times to thoroughly purify acrylamide before use. Afterwards, 2.4 g of acrylamide and 4 mg of N,N'-methylenebisacrylamide were separately dissolved into 6 mL of pure water. 30 mg of DMTPA crystals were gently ground into villous ones before added into the system, followed by the addition of 8 mg of ammonium persulfate. After 10 min of stirring at room temperature to uniformly disperse the DMTPA crystals, the mixture was poured into a round mould and heated at 60 °C for 20 min. The target hydrogel was taken out from the mould and stored at a humid environment.

### Computation details
The calculated molecular models were extracted from corresponding single-crystal structures with the CCDC numbers of 2299585 (DMTPA) and 2299587 (DMTPA-12 h). HOMO and LUMO electron densities and energy levels of the excited states were calculated by time-dependent density functional theory (TD-DFT) using the B3LYP hybrid functional and the 6−31 + G(d,p) basis set. The TD-DFT calculations were performed within Gaussian 16 program. Meanwhile, the noncovalent interactions (NCI) analysis for DMTPA molecules was conducted through Multiwfn package[51] and VMD using the wave functions calculated by Gaussian 16 program.

### Reporting summary
Further information on research design is available in the Nature Portfolio Reporting Summary linked to this article.

## Data availability
The authors declare that the data supporting the findings of this study are available within the article and its Supplementary Information. Extra data are available from the corresponding authors upon request. The X-ray crystallographic coordinates for structures reported in this study have been deposited at the Cambridge Crystallographic Data Centre (CCDC), under deposition numbers CCDC 2299585 (DMTPA), 2299586 (DMTPA-5 min), 2299587 (DMTPA-12 h), 2299619 (DETPA), 2299620 (DETPA-12 h), 2299594 (DiPTPA), 2299595 (DiPTPA-12 h), 2299598 (DMNDCA), 2299599 (DMNDCA-12 h), 2299600 (MMTPA) and 2299601 (MMTPA-12 h). Copies of the data can be obtained free of charge via https://www.ccdc.cam.ac.uk/structures/. Source data are provided with this paper.

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

## Acknowledgements

This work was financially supported by the National Natural Science Foundation of China (52073172 and U22A20250 awarded to W.Z.Y.), the Natural Science Foundation of Shanghai (20ZR1429400 awarded to W.Z.Y.), and the "Shuguang Program" (20SG11 awarded to W.Z.Y.) cosponsored by Shanghai Education Development Foundation and Shanghai Municipal Education Commission. The authors are grateful for the support for the PL and single-crystal structure measurements by Dr. Ruibin Wang and Dr. Lingling Li at IAC of SJTU, respectively.

## Author contributions

W.Z.Y. conceived the idea and directed the project. W.Z.Y. and Z.Z. designed the experiments. Z.Z. and X.C. synthesised the compounds. Z.Z. performed the experiments with assistance from other authors. Z.Z. and Q.Z. performed the HPLC experiments; Z.Z. and A.L. performed the FTIR microspectroscopy; Z.Z. and T.Z. performed the growth morphol-ogy modelling with BFDH method; Z.Z. and Y.C. prepared the hydrogel; all authors contributed to the discussion of the data. Z.Z. wrote the draft manuscript and W.Z.Y. revised the manuscript.

## Competing interests

The authors declare no competing interests.
