## [Peer Review File · Nature Communications]

Photochromic luminescence of organic crystals arising from subtle molecular rearrangementReviewer #1 (Remarks to the Author):

In this paper, the authors report on the photo-induced luminescence color change of the crystal of dimethyl terephthalate (DMTPA). In contrast to previous photo-responsive luminescence switching materials, the luminescent color change is attributed to subtle molecular rearrangements in the crystals induced by light irradiation. While the observed phenomenon is intriguing, it is noteworthy that crystallization-induced phosphorescence and clustering-triggered emission (CTE) by DMTPA have been previously documented in references 34 and 35. The novelty of this study lies in the luminescent changes induced by light exposure, thus necessitating a thorough elucidation of the underlying mechanisms. The authors have conducted detailed mechanistic analyses. However, it appears that the mechanism governing the luminescent changes induced by light may not be fully elucidated. The authors are recommended to address the following points.

- 1) The authors explain that subtle molecular rearrangement occurs upon exposure to light. However, it is recommended to provide an explanation within the main text as to why such rearrangement of molecules takes place without the occurrence of photochemical reactions.
- 2) The difference in intermolecular distances before and after light irradiation is minimal. Although the authors attribute this phenomenon to "amplification effects from the dense and orderly periodic TSC network," a more detailed explanation based on experiments or theoretical calculations is required to enhance clarity.
- 3) While it has been confirmed that there is a slight change in the overall molecular arrangement within the crystal, it is also plausible to consider a mechanism where luminescent color changes occur due to the generation of clusters in lattice defects that have occurred in certain regions of the crystal. A discussion on this aspect is warranted for a more comprehensive understanding of the mechanism.
- 4) This study appears to involve light irradiation across the entire crystal. It would be beneficial to discuss the results of the selective irradiation of only a portion of the crystal. Understanding whether luminescence changes are confined to the irradiated region or if they propagate across the entire crystal from the irradiated point could provide valuable insights into the mechanism of the luminescent changes.

The following minor points should also be addressed.

- 5) In Fig. 1, it is necessary to use superscripts for the numbers 1 and 2 in R1 and R2.
- 6) It is essential to include detailed information about the light irradiation experiments, such as the intensity of light, the apparatus used, and the quantity of the samples, in the Methods section for comprehensive understanding.

Reviewer #2 (Remarks to the Author):

The manuscript reports a photo induced luminescent change from organic crystals. Detail experimental and theoretical investigations suggested that the change was attribute to the generation of clusters which come from the subtle molecular rearrangement induced by light irradiation. While the work presents interesting phenomena and results, the underlying origin of the changes in luminescence and structure remains unclear. Form my perspective, this work may warrant publication in Nature Communications after some issues addressed.

1. I do not consider that the word of "Photo-luminochromism" was suitable to describe the observed phenomenon, and it cannot fully express the meaning of this work.
2. How about investigating the absorption and PL of the DMTPA molecule in solution?
3. The power of the irradiation light is a crucial factor in the photo-triggered process. The authors should provide detailed information about the power of light used in the experiments. Without this information, it is unclear whether lower power can activate the process, and the threshold power for the photo-induced photoluminescence color change needs clarification.
4. In fig.2a, the last time marked on the figure is 500s, but the time marked in the text and Fig.2b, Fig.2c is 1000s, which is inconsistent.
5. In fig.S21, why the absorption band of DMTPA crystals and after 5 min irradiation around 350 nm lower than the baseline of the absorption band of long wavelength? The measurement should be checked carefully.

6. In fig. 2d, the unit of the wavelength will should be annotated.
7. The authors propose a tandem energy transfer process occurring within various clusters, such as clusters 1 and 2. However, more evidence about the formation of these clusters is needed. Are there possibilities for energy transfer from the monomer to cluster 2? The authors should reconsider whether the tandem energy transfer occurred during the excitation process, as the mechanism of long-wavelength emission remains unclear.
8. Regarding the statement "The exposure time under 312 nm UV light was extended to 12 h to amplify the variations detectable," the choice of a 12-hour time scale needs justification. The authors should clarify whether the phenomenon remains consistent after 12 hours.
9. What happens in the crystals during the irradiation process? The molecular rearrangement triggered by light irradiation in these crystals is intriguing. Structural analysis suggests an increase in tightness and enhanced intermolecular interactions. However, the intrinsic driving force remains unclear. If the decrease in bond length and enhanced through-space interactions are the main reasons for the photoluminescence changes, are there other stimuli that can trigger this process?
10. Should interactions other than π - π interactions be considered when exploring intermolecular interactions?
11. In fig. 6b, the emission band below 340 nm were absent.
12. In the results section, it is mentioned "Importantly, this emitting property can persist for more than 3 months under ambient conditions, implying outstanding stability of the newly generated emissive centers." Whether there is some evidence to prove that the emitting property can persist for more than 3 months?
13. In the application section, whether the DMTPA / PAM hydrogel was tested for cyclic stability?

Reviewer #3 (Remarks to the Author):

The paper titled "Photo-luminochromism of organic crystals arising from subtle molecular rearrangement" presents a remarkable exploration into a class of compounds exhibiting photoinduced photoluminescent transformations. These compounds have potential application in information encryption, information storage and biotechnology. This very interesting investigation reports a series of stable crystalline materials that present a broad spectrum of photoluminescence colour variations (p.e. ranging from dark blue to bright orange), while retaining the capacity to revert their initial characteristics.

Additionally, the authors provide a comprehensive analysis of the crystallographic, molecular and electronic properties of the reported molecular systems. Also, this paper presents a meticulous discussion and justification for the noteworthy photoluminescent behaviour of the presented molecules. Finally, the authors present and discuss two showcase examples of application of the reported compounds in information encryption.

Considering the significant contribution and advances presented in this paper concerning smart luminescent materials, and given the quality of the investigation described, I recommend the acceptance of this paper for publication in Nature Communications, after a minor revision is performed.

After, an English revision, I suggest the authors to include an explanation for the statement present in lines 36-38.

The Raman spectra presented in Figure 3, in particular their fingerprint region, are not clearly visible. Hence, I would suggest the authors to include the whole range of the spectra as a clearly visible single figure in the Supplementary Information. If possible, the same should be performed for every FTIR spectra.

The FTIR spectra presented in Figures 3, 4 and 5 do not seem to be normalized. For a better comparison between the spectra, the normalization of the absorbance must be performed.

Finally, in line 371, I would suggest the authors to replace the title of the section from "Discussion" to "Conclusion".

Ms No.: NCOMMS-23-51350A

Ms Title: Photoluminochromism of organic crystals arising from subtle molecular rearrangement

Ms Authors: Zihao Zhao, Yusong Cai, Qiang Zhang, Anze Li, Tianwen Zhu, Xiaohong Chen & Wang Zhang Yuan*

Response to the Comments and Suggestions of Reviewer #1

In this paper, the authors report on the photo-induced luminescence color change of the crystal of dimethyl terephthalate (DMTPA). In contrast to previous photo-responsive luminescence switching materials, the luminescent color change is attributed to subtle molecular rearrangements in the crystals induced by light irradiation. While the observed phenomenon is intriguing, it is noteworthy that crystallization-induced phosphorescence and clustering-triggered emission (CTE) by DMTPA have been previously documented in references 34 and 35. The novelty of this study lies in the luminescent changes induced by light exposure, thus necessitating a thorough elucidation of the underlying mechanisms. The authors have conducted detailed mechanistic analyses. However, it appears that the mechanism governing the luminescent changes induced by light may not be fully elucidated. The authors are recommended to address the following points.

Many thanks for the reviewer's comments to this article.

1. The authors explain that subtle molecular rearrangement occurs upon exposure to light. However, it is recommended to provide an explanation within the main text as to why such rearrangement of molecules takes place without the occurrence of photochemical reactions.

Thanks for the reviewer's advice.

Dimethyl terephthalate (DMTPA) is a relatively stable compound. Without other chemicals, the hydrolysis of ester groups and photolysis that generates cyclohexadienyl radicals are the most possible two photochemical reactions for it. However, the photoinduced photoluminescence (PL) transformation occurs in crystal state, where the lack of hydrogen donor (aqueous or alcoholic media) prohibits these two photochemical reactions (*J. Am. Chem. Soc.* **1992**, *114*, 10461). The impact from water in the air is also excluded by the same PL transformation observed in vacuum.

On the other hand, there are also literatures reporting molecular motions happen in single crystals under UV irradiation can lead to huge fluorescence/phosphorescence enhancement, while no photochemical reactions are involved (*J. Am. Chem. Soc.* **2009**, *131*, 8163; *Nat. Commun.* **2018**, *9*, 840). In excited states, it is possible for aggregates to form enhanced π - π interactions (*Nat. Commun.* **2018**, *9*, 840), or to change into conformation with lower energy level for single molecules (*Matter* **2023**, *6*, 1), thus converting to a metastable state with a different aggregation structure in ground state after radiative decay. Therefore, it is also feasible for DMTPA molecules to rearrange through excited states without the presence of photochemical reactions.

We have added the explanations and corresponding references into the main text.

2. The difference in intermolecular distances before and after light irradiation is minimal. Although the authors attribute this phenomenon to "amplification effects from the dense and orderly periodic TSC

network,” a more detailed explanation based on experiments or theoretical calculations is required to enhance clarity.

Thanks for the reviewer’s comments and suggestions.

As a deduction for the reason why such subtle molecular rearrangements can lead to huge variation in PL, the amplification effect from TSC network is proposed. Actually, in the manuscript, to prove the necessity of the crystalline structure in the photoluminescence (PL), we progressively destroyed the lattice structure of DMTPA (Fig. C1a, b). This experiment can help with the explanation of amplification effect since the internal periodic TSC network is also gradually broken. DMTPA crystals were melted and rapidly dropped into liquid nitrogen for quenching, and then further ground into powders. Analyzed by XRD patterns (Fig. C1a), although remarkably weakened, diffraction peaks of the liquid nitrogen quenched DMTPA solid are still sharp. Even though the quenching extremely constrains the crystallization process, the lattice structure as well as the dense periodic TSC network are retained, endowing the transformation of PL under 312 nm UV irradiation, of which, however, the colour range and response time become narrower and longer (Fig. C1b). As the quenched solid is further ground into powders, the through-space electron interaction network in lattice is severely damaged, thus both the diffraction peaks and the PLC phenomenon becomes neglectable. This experiment suggests the significance of the crystalline nature and the consequent dense and orderly periodic TSC network in PL transformation. Moreover, we have supplemented luminescent photographs of ground powders placed in vacuum (Fig. C1b), where no obvious PLC can be noticed, indicating the inhibited PLC should be ascribed to the attenuation of amplification effect, rather than oxygen quenching to triplet excitons caused by lattice destruction.

Fig. C1. (a) XRD patterns of the crystals, liquid nitrogen quenched solid and ground powder of DMTPA. (b) Luminescent photographs of liquid nitrogen quenched DMTPA solid and its ground powder at ambient and in vacuum under 312 nm UV irradiation with different exposure time. (c) Schematic diagrams of amplification effect in DMTPA and DiPTPA crystals.

Significantly, it is not crystalline structures, but dense and periodic TSC networks (usually resulting from compact molecular packing in lattice structure) that are the key to the amplification effect. The comparison of DMTPA and DiPTPA provide further evidence for this. Condense TSC network among DMTPA molecules enables the domino-like transmission of subtle molecular rearrangements and thus amplify the PL transformation. However, without dense through-space electron interaction network (e.g. π - π interactions) due to the large steric hinderance, though in crystalline state, DiPTPA cannot perform PLC even if molecular rearrangements have been prompted by UV irradiation (Fig. C1c). Working as a “medium” among molecules in the lattice, the TSC network allows the rearrangement of initial molecules to be passed to others, thereby changing the electronic states of the system and generating long-wavelength emission.

Moreover, the proposed amplification effect gains support from the propagation of PL transformation across the entire crystal from the irradiated point (please see the response to Query 4) and a recently published literature, reporting the “domino effect” enabled by through-space conjugation (*J. Am. Chem. Soc.* **2024**, *146*, 2604), which is now cited as Reference 44 in the main text.

The luminescent photographs of ground powders in vacuum have been added into Supplementary Fig. 37. And the additional statements for proving the amplification effect have been added in Supplementary Note 6 (previous Supplementary Note 3) and the main text.

3. While it has been confirmed that there is a slight change in the overall molecular arrangement within the crystal, it is also plausible to consider a mechanism where luminescent color changes occur due to the generation of clusters in lattice defects that have occurred in certain regions of the crystal. A discussion on this aspect is warranted for a more comprehensive understanding of the mechanism.

Thanks for the reviewer’s comments.

According to the reviewer’s suggestion, from two aspects we designed and performed experiments which respectively exclude the possible impacts of macro- and microscopic defects. Corresponding results have been supplemented into the main text and Supplementary Information.

Macroscopic defects such as fluid inclusions, cavities and cracks are firstly taken into consideration. If the luminescent colour changes are associated with these defects, after 312 nm UV irradiation, the changed PL of different regions of the crystal (with or without defects) should be disparate. However, the orange emission is uniformly observed from different parts of DMTPA crystals, no matter those with macroscopic defects or not (Fig. C2a, b), indicating the luminescence colour change does not originate from the macroscopic defects.

For microscopic defects, according to the previous literature, the concentration of permanent lattice defects is finite, leading to the saturated PL intensity at high excitation power density (*ACS Energy Lett.* **2018**, *3*, 1443; *Sci. Rep.* **2013**, *3*, 2657). Similarly, if the emissive centres are clusters generated in lattice defects, their quantity and concentration should be constant, equals to the corresponding lattice defects. In that case, the criteria in the abovementioned literatures are still applicable. The PL intensity ($\lambda_{em} = 650$ and 725 nm) of DMTPA after 312 nm UV treatment demonstrates a linear dependence on the excitation power density (Fig. C2c), indicating the origin of these peaks are photogenerated excitons (*Nat. Commun.* **2022**, *13*, 5712). Therefore, the generation of clusters in microscopic defects is not supposed to be the reason of the photoluminochromism in our work. These literatures are now cited as Reference 36-38 in the main text.

Additionally, with similar molecular packing mode, lattice defects should also be widespread in TPA powder crystals, which, however, show scarcely any change in PL after UV treatment. This result also helps exclude the possibility that the PL transformation originates from the generation of clusters in lattice defects.

Fig. C2. The photograph of DMTPA crystals under (a) bright field and (b) 312 nm UV irradiation. The PL of DMTPA crystals have transformed after UV treatment. (c) The plots of PL intensity ($\lambda_{em} = 650$ and 725 nm) as a function of excitation power density ($\lambda_{ex} = 312$ nm).

The plots of PL intensity ($\lambda_{em} = 650$ and 725 nm) as a function of excitation power density ($\lambda_{ex} = 312$ nm) have been supplemented into Supplementary Information as Supplementary Fig. 26, and the corresponding statements have also been added in Supplementary Note 3 and the main text.

4. This study appears to involve light irradiation across the entire crystal. It would be beneficial to discuss the results of the selective irradiation of only a portion of the crystal. Understanding whether luminescence changes are confined to the irradiated region or if they propagate across the entire crystal from the irradiated point could provide valuable insights into the mechanism of the luminescent changes.

Many thanks for the reviewer's reminder and advice. This does be able to provide new insights for the mechanism discussion.

Corresponding experiments were designed and performed. Part of the bulk DMTPA crystal was wrapped in aluminium foil to block the 312 nm UV irradiation (Fig. C3). After 10 min of irradiation, the part exposed to UV irradiation demonstrated orange PL while the covered area, after removing the aluminium foil, still retained the blue emission. It seems that the luminescence changes are only confined to the irradiated region. However, on closer inspection, a small region at the boundary of the irradiated and covered part demonstrated purple PL, which is exact the intermediate state of the PL transformation process. To investigate whether this phenomenon only happens at the intersection or can be propagated to the deeper site of the covered part of crystal, we prolonged the irradiation time to facilitate the completion of possible propagation process. After 1 h of irradiation and removal of the aluminium foil, the covered region was completely turn into purple emission. These results indicate the PL colour transformation process happens at the irradiated region in a short time, while can be propagated across the crystal in the long run. This could also provide extra evidence for the proposed amplification effect, which, with the TSC network acting like a "medium" throughout the lattice structure, enables the propagation of PL transformation across the entire crystal.

As a modification, Fig. C3 and corresponding discussion have been added into the main text and Supplementary Information as Supplementary Fig. 32 and Supplementary Note 5, respectively.

Fig. C3. Luminescent photographs of DMTPA crystals after locally irradiated by 312 nm UV light with different exposure time.

The following minor points should also be addressed.

5. In Fig. 1, it is necessary to use superscripts for the numbers 1 and 2 in R1 and R2.

Many thanks for the reviewer's reminder and sorry for the mistake.

The numbers 1 and 2 in R1 and R2 have been changed to superscripts.

6. It is essential to include detailed information about the light irradiation experiments, such as the intensity of light, the apparatus used, and the quantity of the samples, in the Methods section for comprehensive understanding.

Thanks for the reviewer's advice.

The detailed information of the excitation light source and the quantity of the samples during the photoluminochromic process has now been supplemented in the Methods section with the subtitle "Irradiation experiments".

Response to the Comments and Suggestions of Reviewer #2

The manuscript reports a photo induced luminescent change from organic crystals. Detail experimental and theoretical investigations suggested that the change was attribute to the generation of clusters which come from the subtle molecular rearrangement induced by light irradiation. While the work presents interesting phenomena and results, the underlying origin of the changes in luminescence and structure remains unclear. Form my perspective, this work may warrant publication in Nature Communications after some issues addressed.

Many thanks for the reviewer's comments and appreciation of our work.

1. I do not consider that the word of "Photo-luminochromism" was suitable to describe the observed phenomenon, and it cannot fully express the meaning of this work.

Thanks for the reviewer's comments.

Initially, we contemplated adopting established terms such as "photochromism" or "photoswitch" to encapsulate the photoinduced photoluminescence (PL) transformation observed in our research. Nevertheless, "photochromism" typically describes the colour change in substances in reaction to

photostimuli, and "photoswitch" denotes molecules that alter their structural geometry and chemical characteristics upon irradiation. In contrast, DMTPA and its derivatives manifest distinct attributes, including the maintenance of colour appearance during PL transformation, the emergence of new emission peaks rather than merely enhancing existing ones, and a PL variation propelled by a purely physical process. Given the scarcity of similar research, devising a universal term that encapsulates all these traits proves challenging. Hence, we opted to coin a new term for the overarching category that includes all materials undergoing photoinduced PL transformation, prioritizing descriptive accuracy over precision.

Previous literature employs terms like "mechanoluminochromism" or "mechanochromic luminescence" and "electroluminochromism" to describe changes in intensity and/or colour of photoluminescent materials induced by mechanical forces or electrically driven redox reactions, respectively (*Angew. Chem. Int. Ed.* **2016**, *55*, 7998; *J. Am. Chem. Soc.* **2010**, *132*, 2160; *Adv. Mater.* **2022**, *34*, 2107013). Drawing inspiration from these precedents, we propose "photoluminochromism" to denote the variations in intensity and/or colour of photoluminescence triggered by external photostimuli. The prefix "photo" signifies that the discoloration of PL is initiated by light exposure, while "lumino" indicates that the colour change pertains to material emission. This term encompasses a broad spectrum of research, including molecular photoswitches, photoactivated phosphorescence, photoinduced radicals, and other photoresponsive luminescent materials. Consequently, we advocate for the retention of "photoluminochromism" in our manuscript. Nonetheless, we are open and receptive to adopting a more precise term should one be proposed.

2. How about investigating the absorption and PL of the DMTPA molecule in solution?

Thanks for the reviewer's advice. We have added corresponding experimental results in the main text and Supplementary Information.

The UV-vis absorption of 10^{-4} M DMTPA/2-MTHF solution exhibits an evident E₂ band (240 nm) and a B band (285 nm) of benzene ring (Fig. C4a). Except for the Raman peak at 345 nm, the concentrated (10^{-1} M) DMTPA/2-MTHF solution demonstrates emission centred at 374 nm with considerable PL intensity, which drastically attenuates as the concentration decreases to 10^{-2} M and lower, illustrating a concentration-enhanced emission characteristic (Fig. C4b). According to the clustering-triggered emission (CTE) mechanism, DMTPA molecules are discrete in dilute solutions, while efficiently clusters with each other in concentrated counterparts. Therefore, in concentrated solutions, the original limited molecular conjugation is effectively extended, in favour of emission, and the severe nonradiative deactivations can be restrained. Furthermore, when dissolved in different solvents, DMTPA solutions (10^{-3} M) show diverse emission spectra (Fig. C4c). For instance, the DMTPA/DMF solution exhibits an intense emission peak centred at 415 nm, while the DMTPA/EA solution shows two inconspicuous peaks at 364 and 374 nm. Various polarity and solubility endow DMTPA shows different degrees of dispersion in these solvents, leading to distinct emission spectra. Although these peaks seem disorganized, most of them can be found in emission spectrum of 10^{-1} M DMTPA/2-MTHF solution. The concurrence of these emission bands in concentrated solution indicates the presence of diverse clusters in aggregated states.

Fig. C4. (a) Absorption of 10^{-4} M DMTPA/2-MTHF solution. Emission spectra of (b) varying DMTPA/2-MTHF solutions and (c) DMTPA solutions (10^{-3} M) with different solvents ($\lambda_{\text{ex}} = 312$ nm).

We have placed these data into the revised Supplementary Information and added corresponding discussion in the revised manuscript.

3. The power of the irradiation light is a crucial factor in the photo-triggered process. The authors should provide detailed information about the power of light used in the experiments. Without this information, it is unclear whether lower power can activate the process, and the threshold power for the photo-induced photoluminescence color change needs clarification.

Thanks for the reviewer's comments.

The detailed information about the excitation light source has been supplemented in the Methods section with the subtitle "Irradiation experiments". 312 nm UV irradiations of different power densities from 0.06 to 4.16 mW/cm² are all capable of prompting the PL colour transformation. However, the process is slowed down as the power density decreases. Uniformly irradiated for 2 min, the PL colour of DMTPA crystals changes to orange-yellow under a relatively high power density of 4.16 mW/cm², while only changes to dark purple with a low power density of 0.06 mW/cm² (Fig. C5a). Corresponding emission spectra of DMTPA crystals after 2 min of 312 nm UV irradiation with diverse power densities show characteristics after PL transformation (Fig. C5b). These results indicate the equivalency of exposure time and power density.

Fig. C5. (a) Luminescent photographs and (b) emission spectra of DMTPA crystals after 2 min of 312 nm UV irradiation with different power densities ($\lambda_{\text{ex}} = 312$ nm).

4. In fig.2a, the last time marked on the figure is 500s, but the time marked in the text and Fig.2b, Fig2c is 1000s, which is inconsistent.

Thanks for the reviewer's reminder and sorry for the mistake.

The luminescent photograph of DMTPA after 1000 s of 312 nm UV irradiation is supplemented in Fig. 2a.

5. In fig.S21, why the absorption band of DMTPA crystals and after 5 min irradiation around 350 nm lower than the baseline of the absorption band of long wavelength? The measurement should be checked carefully.

Many thanks for the reviewer's reminder.

The absorption of DMTPA crystals was re-measured and has been added into the Supplementary Information.

6. In fig.2d, the unit of the wavelength will should be annotated.

Thanks for the reviewer's reminder and sorry for the mistake.

The unit of the emission wavelength has now been annotated in Fig. 2d.

7. The authors propose a tandem energy transfer process occurring within various clusters, such as clusters 1 and 2. However, more evidence about the formation of these clusters is needed. Are there possibilities for energy transfer from the monomer to cluster 2? The authors should reconsider whether the tandem energy transfer occurred during the excitation process, as the mechanism of long-wavelength emission remains unclear.

Thanks for the reviewer's comments.

The PL transformation of DMTPA crystals under 312 nm UV irradiation can be concluded as two parts, including the generation of long-wavelength peaks as well as the decline of peak at ~420 nm (Fig. C6a). The former is ascribed to the new emissive aggregates/clusters created by the photoinduced subtle molecular rearrangement, while the latter is explained by the absorption of newly generated emissive clusters, namely the proposed energy transfer process. The main text has been revised for a clearer statement. Below please find the detailed discussion to each issue.

(1) About the evidence of the formation of these clusters. The formation process of the clusters (emissive centres) can be reflected by the newly generated long-wavelength emission peaks. In the analysis of the emission spectra of dilute (10^{-5} M) DMTPA/2-MTHF solution at 77 K in the main text, we have attributed the peaks at 302 and 420 nm to the fluorescence and phosphorescence for monomeric DMTPA, respectively. Therefore, photogenerated peaks (515/550/595/650/725 nm), for their longer wavelengths than the monomeric ones, can only originate from different clustered DMTPA aggregates formed during UV treatment, according to the Kasha's rule and different lifetimes of these peaks (Fig. C6b). Moreover, the formation of new emissive clusters is also proved by the newly emerging absorption peak at ~350 nm, indicating the presence of emissive components with larger conjugation in DMTPA crystals after 312 nm UV irradiation (Fig. C6c).

(2) The energy transfer from the monomer to "cluster 2" is infeasible, because the excitation spectrum of the long-wavelength phosphorescence ($\lambda_{em} = 725$ nm) exhibit no peak around ~420 nm. Moreover, the emission spectrum with $\lambda_{ex} = 420$ nm (Fig. C6d) do not show emission peak around 725 nm. Thus, the monomer emission could not be utilized by "cluster 2".

By contrast, the “cluster 1” here refers to two kinds of clusters with intrinsic phosphorescence at 595 and 650 nm, respectively, while the “cluster 2” refers to the ones with phosphorescence at 725 nm. According to the excitation spectra of DMTPA crystals after 312 nm UV irradiation (Fig. C6e), there is a broad excitation band including a peak at ~ 420 nm with emission wavelength of 595 and 650 nm. Therefore, once the “cluster 1” is formed, the energy from triplet excitons of DMTPA monomers can be transferred to it. For “cluster 2” with emission wavelength of 725 nm, the excitation spectrum only exhibits one peak at exactly 650 nm, corresponding to the “cluster 1” excitons. Thus the energy transfer from the monomer to “cluster 2” cannot directly happen, while the one from “cluster 1” to “cluster 2” is possible.

Fig. C6. (a) Prompt emission spectra of DMTPA crystals under 312 nm UV irradiation with different exposure time. (b) Millisecond (ms)-scale lifetimes for DMTPA crystals after 1000 s of 312 nm UV irradiation ($\lambda_{ex} = 312$ nm). (c) Absorption of DMTPA crystals before and after 5 min and 12 h of 312 nm UV irradiation. (d) Emission spectrum and corresponding multi-peaks Gaussian fitting of DMTPA crystals after 1000 s of 312 nm UV irradiation ($\lambda_{ex} = 420$ nm). (e) Excitation spectra of DMTPA crystals with varied emission wavelengths. (f) Schematic illustration of proposed tandem energy transfer process. (g) Ms-scale lifetimes for DMTPA crystals before UV irradiation ($\lambda_{ex} = 312$ nm).

(3) As abovementioned, the long-wavelength emission peaks should derive from the formation of new emissive centres. However, the generation of new emissive clusters does not lead to the decline of monomeric phosphorescence. Therefore, some sort of nonradiative decay should occur on the monomeric phosphorescence. Considering the scarcely changed rigid crystalline environment versus such large alteration of emission, vibrational and rotational dissipation of molecules should not be the main cause. In

that case, possible energy transfer process is taken into account (Fig. C6f). The consistency between wavelength of monomeric phosphorescence and excitation of clusters inspired us the feasibility of the first stage of energy transfer (monomer \rightarrow “cluster 1”). Moreover, the shortened millisecond-scale lifetime at 420 nm after PL transformation also validates this energy transfer process (Fig. C6b, g). Up to this point, the decline of peak at \sim 420 nm has already been well rationalized. However, the excitation peak at 650 nm of “cluster 2” reminds certain possibility of subsequent energy flow (“cluster 1” \rightarrow “cluster 2”), which is also included as a potential successive stage of energy transfer process.

8. Regarding the statement "The exposure time under 312 nm UV light was extended to 12 h to amplify the variations detectable," the choice of a 12-hour time scale needs justification. The authors should clarify whether the phenomenon remains consistent after 12 hours.

Many thanks for the reviewer’s comments and sorry for the ambiguity of discussion.

In addition to the transformation from dark blue to bright orange in the first 15 min, such PL colour variation continues over a longer irradiation period, albeit at a much slower pace and with a more limited scope of change. As shown in Fig. C7a, the PL colour gradually changes into tangerine after 5 h of 312 nm UV irradiation and gets deep red when the exposure time reaches 12 h. The PL of DMTPA crystals is basically stable after 12 h of irradiation, which can be proved by the scarcely changed emission spectra (Fig. C7b). Therefore, the transformation process largely stopped and the phenomenon remains consistent after 12 h of irradiation. With the aiming of accumulating the structural variations during the PL transforming period as much as possible, the 12-hour time scale was chosen.

Fig. C7. (a) Luminescent photographs and (b) emission spectra of DMTPA crystals under 312 nm UV irradiation with different exposure time (> 15 min).

We have added these data into the revised manuscript or Supplementary Information and discussed them in the revised manuscript.

9. What happens in the crystals during the irradiation process? The molecular rearrangement triggered by light irradiation in these crystals is intriguing. Structural analysis suggests an increase in tightness and enhanced intermolecular interactions. However, the intrinsic driving force remains unclear. If the decrease in bond length and enhanced through-space interactions are the main reasons for the photoluminescence changes, are there other stimuli that can trigger this process?

Many thanks for the reviewer's comments. As a preliminarily proposed mechanism, we believe there is still much room for modification and improvement

As reported by the previous literatures, the driving force could be the conformation adjustments for single molecules and/or aggregates occur in excited states. The CS-CF₃ reported by Li et al is supposed to perform molecular motion in excited state owing to the tendency of excited aggregates to form enhanced π - π interactions (*Nat. Commun.* **2018**, 9, 840). It is also reported by Chi et al and Ma et al, respectively, that the conformations of DBBZ and 4-BZ single molecules, whose structure is similar to DMTPA, in excited states tend to adjust depending on the rigidity of the environment, thus resulting in stimuli-responsive emission (*Matter* **2023**, 6, 1; *J. Am. Chem. Soc.* **2023**, 145, 16748). Therefore, it is reasonable to deduce similar molecular rearrangement occurs in excited states of DMTPA aggregates, driven by the different electron distribution and aggregate conformation compared with those in ground state.

To prove such aggregate conformation conversion, we simulated the conformational change of DMTPA trimer at the lowest triplet excited state (T₁). Compared with the S₀ state, the optimized T₁ conformation tends to be more planar, with the distance between the adjacent parallel benzene rings elongating from 6.016 to 6.589 Å while the interplanar spacing shortening from 2.277 to 1.977 Å (Fig. C8a). The phosphorescence energy calculated by TD-DFT is 2.41 eV, which is in good consistency with one of the photogenerated peaks at 515 nm. However, such emission at 515 nm is not observed until exposed to 312 nm UV irradiation (Fig. C6a), indicating the corresponding conformation of DMTPA trimer is modified at excited states by spontaneous molecular rearrangements. Nevertheless, it should be noted that the orange-luminescent state of DMTPA can be retained after the first PL transformation, without repetitive transition during next irradiation. It is reasonable to speculate a metastable state will be arrived after the first radiative decay from T₁ state. With a different conformation from the thermodynamically stable state, the metastable aggregates (photogenerated aggregates) can be directly excited, affording new triplet states as well as corresponding phosphorescence.

Fig. C8. (a) The optimized S₀ and T₁ conformations of DMTPA trimer, as well as the schematic illustration of mechanism. (b) The emission spectrum of tablet of DMTPA crystals ($\lambda_{\text{ex}} = 312$ nm).

Considering the increment in tightness and enhanced intermolecular interactions after 312 nm UV irradiation, we prepared a tablet with DMTPA crystals upon pressurization (550 MPa). As Fig. C8 shows, before exposure to 312 nm UV irradiation, the DMTPA tablet exhibits inconspicuous emission band at ~420 nm and obvious peaks at 595, 650 and 725 nm, as well as a negligible peak at 515 nm. This phenomenon is quite similar to the transformed state of DMTPA crystals after 312 nm UV irradiation. Although the photophysical properties of DMTPA tablet are not totally consistent with those of DMTPA crystals after UV treatment owing to the different rigidity of environment, the main transformation of emissive centres can be triggered by pressurization.

10. Should interactions other than π - π interactions be considered when exploring intermolecular interactions?

Thanks for the reviewer's comments.

Yes, other interactions have been involved in this work. The through-space conjugation (TSC) network is composed by multifarious interactions, thus the role played by each kind of through-space electron interaction should be considered. In DMTPA crystals, the TSC network is mainly formed by the evident π - π interactions among phenyl rings and the short contacts (C=O \cdots H-C and C=O \cdots C) between ester groups. Therefore, the variations of these two kinds of interactions before and after the UV irradiation are both discussed via FTIR, single crystal XRD and growth morphology modeling. Furthermore, the importance of these two interactions in PL transformation is separately proved by the comparison of DMTPA and its derivatives.

Actually, it is not necessarily the π - π interactions, but the condense TSC network formed by any kind of interactions is the key to the amplification effect and consequent PL transformation. The photoinduced PL transformation process of *N,N'*-carbonylbis succinimide (CBSI) without π - π interaction supports this conclusion (Fig. C9), which is going to be reported in our subsequent work.

Fig. C9. Luminescent photographs of CBSI crystals (a) before and (b) after 365 nm UV irradiation ($\lambda_{\text{ex}} = 312$ nm).

11. In fig. 6b, the emission band below 340 nm were absent.

Many thanks for the reviewer's reminder and sorry for this mistake.

In order to avoid the influence of excitation wavelength, we had to start to collect signals from a few tens of nanometers behind the excitation wavelength (312 nm). Therefore, the signal of the short wavelength peak could not be collected completely. To show the emission signal excited at 312 nm as completely as possible, we have adjusted the starting abscissa of the emission spectrum to 330 nm, as the updated Fig. 6b shows.

12. In the results section, it is mentioned "Importantly, this emitting property can persist for more than 3 months under ambient conditions, implying outstanding stability of the newly generated emissive centers." Whether there is some evidence to prove that the emitting property can persist for more than 3 months?

Thanks for the reviewer's comments.

The luminescent photographs of DMTPA crystals placed under ambient conditions for different time demonstrate a gradually reversing tendency of the PL transformation (Fig. C10a). The DMTPA crystals were irradiated under 312 nm UV light for 15 min to proceed the PL transformation. After 2 weeks of placement, the transformed DMTPA crystals still demonstrated bright orange emission, which gradually faded as the time increases to 1.5 months. For a part of crystals, the PL became purple after 3 months placement, while most of them still maintained light orange emission. When the placing time reaches 5 months, most of DMTPA crystals resumed purple emission, indicating the spontaneous reversibility. Corresponding emission spectra display the same experimental results (Fig. C10b).

Fig. C10. (a) Luminescent photographs and (b) emission spectra of DMTPA crystals placed under ambient conditions for different time ($\lambda_{\text{ex}} = 312 \text{ nm}$).

We have added these data into the revised Supplementary Information as Supplementary Fig. 15.

13. In the application section, whether the DMTPA / PAM hydrogel was tested for cyclic stability?

Thanks for the reviewer's comment.

The cyclic stability of the PL transformation process of DMTPA/PAM hydrogel is tested and supplemented into the Supplementary Information as Supplementary Fig. 65. Using the ratio of PL intensities at 650 and 420 nm (I_{650}/I_{420}) as the indicator, this PL transformation and restoring process is proved can be repeated for 10 cycles (Fig. C11).

Fig. C11. Plot of luminescence intensity ratio versus repeated phototriggering and heating cycles. I_{650} and I_{420} refer to the PL intensities at 650 and 420 nm, respectively ($\lambda_{\text{ex}} = 312$ nm).

Response to the Comments and Suggestions of Reviewer #3

The paper titled “Photo-luminochromism of organic crystals arising from subtle molecular rearrangement” presents a remarkable exploration into a class of compounds exhibiting photoinduced photoluminescent transformations. These compounds have potential application in information encryption, information storage and biotechnology. This very interesting investigation reports a series of stable crystalline materials that present a broad spectrum of photoluminescence colour variations (p.e. ranging from dark blue to bright orange), while retaining the capacity to revert their initial characteristics.

Additionally, the authors provide a comprehensive analysis of the crystallographic, molecular and electronic properties of the reported molecular systems. Also, this paper presents a meticulous discussion and justification for the noteworthy photoluminescent behaviour of the presented molecules. Finally, the authors present and discuss two showcase examples of application of the reported compounds in information encryption.

Considering the significant contribution and advances presented in this paper concerning smart luminescent materials, and given the quality of the investigation described, I recommend the acceptance of this paper for publication in Nature Communications, after a minor revision is performed.

Many thanks for the reviewer’s comments and appreciation of our work.

After, an English revision, I suggest the authors to include an explanation for the statement present in lines 36-38.

Thanks for the reviewer’s advice.

The language of this paper has been carefully polished.

Moreover, the explanation for the statement “nevertheless, most of them, especially those undergone photochemical reactions, experience colour change of appearance during photoluminescence (PL) transformation, which limits their applications in information encryption” has been included into the main text, right after the above sentence. The supplemented explanation is shown below:

Namely, once input, the information can be directly read without decoding (e.g. photoexcitation), for the colour of materials in corresponding regions will change.

The Raman spectra presented in Figure 3, in particular their fingerprint region, are not clearly visible. Hence, I would suggest the authors to include the whole range of the spectra as a clearly visible single figure in the Supplementary Information. If possible, the same should be performed for every FTIR spectra.

Thanks for the reviewer's advice.

The whole range of the Raman spectra and every FTIR spectra have now been supplied in the Supplementary Information.

The FTIR spectra presented in Figures 3, 4 and 5 do not seem to be normalized. For a better comparison between the spectra, the normalization of the absorbance must be performed.

Many thanks for the reviewer's suggestion.

As the modification, the FTIR spectra in the manuscript and Supplementary Information have all been replaced by the normalized ones. For clarity, only the region corresponding to the skeletal vibrations of aromatic ring are shown in the main text, while the spectra containing the reference peak for normalization are exhibited in Supplementary Information.

Finally, in line 371, I would suggest the authors to replace the title of the section from "Discussion" to "Conclusion".

Many thanks for the reviewer's suggestion. However, we noticed that the conclusion part is usually entitled as "Discussion" in recent publications and author guideline in Nature Communications. Therefore, we would like preserve the title "Discussion". Still, we appreciate the reviewer's kind advice.

We are now sending our revised manuscript to your editorial office *via* the online submission system. Thank you and the reviewers very much for your review and consideration. We appreciate your spending precious time to handle our manuscript. We look forward to hearing from you.

Yours sincerely,

W. Z. Yuan Ph.D.

Reviewer #1 (Remarks to the Author):

The manuscript has been improved significantly according to the comments by the reviewers. However, the following points have not been fully addressed.

Reviewer 1, comment 1 and Reviewer 2, comment 9: The authors have referenced literatures to explain that the phenomenon of photo-induced alteration of luminescence properties in the crystalline state without photochemical reactions has precedents. However, they do not thoroughly explain how their findings compare to previous studies. For instance, in J. Am. Chem. Soc. 2009, 131, 8163, it appears that photochemical [2 + 2] cycloaddition contributes to the increase in luminescence intensity upon light irradiation. Additionally, Matter 2023, 6, 1 suggests that changes in molecular alignment due to grinding or thermal annealing can alter the excitation state conformation. Nat. Commun. 2018, 9, 840 presents an example of photo-induced phosphorescence. While the authors cite this article on page 2 of their paper, they do not adequately discuss its relevance as a precedent for increased luminescent properties in crystals upon light irradiation. Therefore, the authors need to clarify how their work, in comparison to the past example, possesses the novelty and significance worthy of a publication in the prestigious journal Nat. Commun.

Reviewer 2, comment 1: The authors propose the use of the term "photoluminochromism" to describe the phenomenon where light irradiation leads to a change in the emission color. However, it should be noted that there are multiple precedents in the literature where this phenomenon has been referred to as "photochromic luminescence." Considering these precedents, it may be more appropriate to use the established terminology of "photochromic luminescence" to describe this phenomenon. The authors are also recommended to cite precedents of "photochromic luminescence".

Reviewer #2 (Remarks to the Author):

The authors have responded the comments carefully. Yet, there are still some points should be addressed.

1.Regarding question 1, my opinion is that the authors could use more precise words and phrases to describe the aspects and significance of this work. It's not necessary to coin a new word to encapsulate this phenomenon. Both "photo" and "lumino" inherently imply light, which could easily confuse readers. The author might consider using clearer terminology to describe this phenomenon, such like photoinduced luminochromism.

2.Regarding question 2, based on the emission spectra at different concentrations, it is possible that DMTPA forms excimers. How does the author confirm whether the long-wavelength emission originates from excimers or the CTE mechanism in solution? On the other hand, in the spectral tests and analyses in different solvents, the author uses a concentration of 10^{-3} M for testing. This concentration may cause some aggregation or nano-precipitation due to the solubility of the molecules, which often affecting the results. It is recommended that the author uses an extremely dilute solution in 10^{-5} M to measure the spectra in different solvents. Additionally, two emission peaks occur in different solvents, please provide a more detailed explanation about the attribution of these two peaks, where the peak at 364 nm keep the same position at various solutions.

3.Regarding question 3, the authors claim that the exposure time and powder density are equivalent. Therefore, as I understand it, under very low light intensity, far below the threshold intensity (the author did not provide data related to the threshold mentioned in the previous question), it is possible to achieve color change by irradiating for a long period of time? Could the author provide a more reasonable explanation and corresponding data?

4.Regarding question 7. could the authors provide some direct evidence to prove the formation of clusters? Moreover, from the excitation spectra of 595 and 650 nm, are there more forms of clusters present?

Reviewer #3 (Remarks to the Author):

The authors have diligently addressed the reviewers' comments, including my own, thereby strengthening the manuscript. Consequently, I recommend its publication in Nature Communications.

Ms No.: NCOMMS-23-51350B

Ms Title: Photochromic luminescence of organic crystals arising from subtle molecular rearrangement

Ms Authors: Zihao Zhao, Yusong Cai, Qiang Zhang, Anze Li, Tianwen Zhu, Xiaohong Chen & Wang Zhang Yuan

Response to the Comments and Suggestions of Reviewer #1

The manuscript has been improved significantly according to the comments by the reviewers. However, the following points have not been fully addressed.

Thanks for the reviewer's comments.

*1. Reviewer 1, comment 1 and Reviewer 2, comment 9: The authors have referenced literatures to explain that the phenomenon of photo-induced alteration of luminescence properties in the crystalline state without photochemical reactions has precedents. However, they do not thoroughly explain how their findings compare to previous studies. For instance, in *J. Am. Chem. Soc.* 2009, 131, 8163, it appears that photochemical [2 + 2] cycloaddition contributes to the increase in luminescence intensity upon light irradiation. Additionally, *Matter* 2023, 6, 1 suggests that changes in molecular alignment due to grinding or thermal annealing can alter the excitation state conformation. *Nat. Commun.* 2018, 9, 840 presents an example of photo-induced phosphorescence. While the authors cite this article on page 2 of their paper, they do not adequately discuss its relevance as a precedent for increased luminescent properties in crystals upon light irradiation. Therefore, the authors need to clarify how their work, in comparison to the past example, possesses the novelty and significance worthy of a publication in the prestigious journal *Nat. Commun.**

Thanks for the reviewer's comments and suggestion.

In most examples without photochemical reactions, the alteration of photoluminescence (PL) properties often arises from intramolecular conformational adjustment triggered by external stimuli, like the change of dihedral angle in benzil group (*Matter* 2023, 6, 1).

In our case, however, the remarkable photochromic luminescence (PCL) stems from the change of molecular aggregation state. Specifically, it is associated with the emergence of new emissive clusters, occurring without notable change at the individual molecular level. Meanwhile, despite several instances of PL change have been documented without intramolecular conformational adjustment, they are confined to the PL enhancement of already existing species (*J. Am. Chem. Soc.* 2009, 131, 8163; *Nat. Commun.* 2018, 9, 840), and no dynamic evolution of emissive species is involved. In contrast, emerging emissive clusters are formed in DMTPA crystals upon UV irradiation, leading to the drastic PCL from blue to red. Moreover, such red emission is rarely found in single-benzene based structures.

Following the reviewer's suggestion, additional discussion have been included in the revised manuscript.

2. Reviewer 2, comment 1: The authors propose the use of the term "photoluminochromism" to describe the phenomenon where light irradiation leads to a change in the emission color. However, it should be noted that there are multiple precedents in the literature where this phenomenon has been referred to as "photochromic luminescence." Considering these precedents, it may be more appropriate to use the

established terminology of "photochromic luminescence" to describe this phenomenon. The authors are also recommended to cite precedents of "photochromic luminescence".

Thanks for the reviewer's comments and suggestions.

Following the reviewers' and editor's suggestion, we have changed the original "photoluminochromism" into the term of "photochromic luminescence". Meanwhile, several precedents of "photochromic luminescence" (*Angew. Chem. Int. Ed.* **60**, 11247 (2021); *J. Am. Chem. Soc.* **145**, 16748 (2023); *J. Am. Chem. Soc.* **131**, 8163 (2009)) have also been cited in the revised manuscript.

Response to the Comments and Suggestions of Reviewer #2

The authors have responded the comments carefully. Yet, there are still some points should be addressed.

Thanks for the reviewer's comments.

1. Regarding question 1, my opinion is that the authors could use more precise words and phrases to describe the aspects and significance of this work. It's not necessary to coin a new word to encapsulate this phenomenon. Both "photo" and "lumino" inherently imply light, which could easily confuse readers. The author might consider using clearer terminology to describe this phenomenon, such like photoinduced luminochromism.

Thanks for the reviewer's comments and suggestion.

Following the common suggestions from reviewers #1 and #2 and also the advice from the editor, we used "photochromic luminescence" to replace "photoluminochromism" in the revised manuscript.

2. Regarding question 2, based on the emission spectra at different concentrations, it is possible that DMTPA forms excimers. How does the author confirm whether the long-wavelength emission originates from excimers or the CTE mechanism in solution? On the other hand, in the spectral tests and analyses in different solvents, the author uses a concentration of 10^{-3} M for testing. This concentration may cause some aggregation or nano-precipitation due to the solubility of the molecules, which often affecting the results. It is recommended that the author uses an extremely dilute solution in 10^{-5} M to measure the spectra in different solvents. Additionally, two emission peaks occur in different solvents, please provide a more detailed explanation about the attribution of these two peaks, where the peak at 364 nm keep the same position at various solutions.

Thanks for the reviewer's comments and suggestion.

For the emission of DMTPA in solution, the monomer emission is at ~305 nm (dependent on the solvent) in 2-MTHF and the excimer emission usually locates at 364 and 374 nm (*Macromolecules*. Pergamon Press, Oxford, 1982, pp. 139; *Macromolecules* **51**, 9035 (2018)). Therefore, other bands beyond the excimer emission are ascribable to the cluster emissions. For example, previously, we have measured the emission spectra of DMTPA in trifluoroacetic acid (TFA) across various concentrations. As shown in Fig. C1, in dilute solutions, the monomer emission is around 332 nm. In highly concentrated solutions (*e.g.* 0.5 M), however, bluish-green emission with maxima at 460/492/534 nm are noticed, on account of the clustering of DMTPA molecules. Therefore, we can generally identify the emission species based on the PL peaks.

Fig. C1. PL properties of DMTPA/TFA solutions at different concentrations. (a) Photograph taken under 312 nm UV irradiation. PL spectra with λ_{ex} s of (b) 280 and (c) 410 nm. Reproduced with permission from *Macromolecules* **51**, 9035 (2018). Copyright 2018 American Chemical Society.

Regarding PL spectra in varying solvents, following the reviewer's suggestion, we newly measured the emission spectra using an extremely dilute solution of 10^{-5} M. As can be seen in Fig. C2a, the fluorescence for monomeric DMTPA locates at ~ 305 nm and slightly varies due to the influence of the solvents. The peaks at 364 and 374 nm for DMTPA excimers are observed in emission spectra of DMTPA/DMF and DMTPA/hexane solutions, while absent in the counterparts of DMTPA/2-MTHF, DMTPA/ CHCl_3 and DMTPA/EtOH solutions. Just as the reviewer pointed out, this can be ascribed to the different solubilities of DMTPA in diverse solvents.

Fig. C2. Normalized emission spectra of (a) 10^{-5} and (b) 10^{-3} M DMTPA in varying solvents.

Indeed, two peaks at 364 and 374 nm are observed, which should be assigned to the excimer emission of DMTPA, occurring at a concentration of 10^{-3} M in different solvents (Fig. C2b). Therefore, these two peaks basically keep the same position in different solvents.

Following the reviewer's suggestion, we have added the new results into the revised Supplementary Information with corresponding discussion in the revised manuscript. The detailed explanation of different peaks is also supplemented into Supplementary Note 2.

3. Regarding question 3, the authors claim that the exposure time and powder density are equivalent. Therefore, as I understand it, under very low light intensity, far below the threshold intensity (the author did not provide data related to the threshold mentioned in the previous question), it is possible to achieve color change by irradiating for a long period of time? Could the author provide a more reasonable explanation and corresponding data?

Thanks for the reviewer's comments and questions.

Even under an extremely weak power density of $5 \mu\text{W}/\text{cm}^2$, which is the minimum detectable level of our equipment, the emission color of DMTPA crystals can undergo a transition from blue to purple after being irradiated with 312 nm UV light for 5 h (Fig. C3). Due to the limitations of our instrumentation, we cannot acquire the accurate threshold, which must be below $5 \mu\text{W}/\text{cm}^2$.

However, considering the fact that emission of DMTPA crystals does not undergo transformation under daylight, despite containing trace amounts of 312 nm UV light, there should be a threshold of UV power density that required to trigger the change in emission colour. Therefore, the equivalency of exposure time and power density can only be established under the condition that the incoming UV stimuli exceed such threshold.

Fig. C3. Luminescent photograph of DMTPA crystals before and after 312 nm UV irradiation (power density = $5 \mu\text{W}/\text{cm}^2$) for 5 h.

4. Regarding question 7. could the authors provide some direct evidence to prove the formation of clusters? Moreover, from the excitation spectra of 595 and 650 nm, are there more forms of clusters present?

Thanks for the reviewer's questions.

Inferred from the PL evolution of DMTPA crystals, clusters with long-wavelength emission far beyond excimer emissions are formed during UV irradiation. Although it is difficult to directly probe the formation process of emissive clusters in crystals, we tried to gain further information on the clustering process in solution, where long-wavelength emissive clusters are formed upon concentration.

The dynamic light scattering (DLS) experiments were carried out on DMTPA/2-MTHF solutions with different concentrations. As shown in Fig. C4, in 10^{-5} M solution, the hydrodynamic diameter of molecularly dispersed DMTPA is around 1 nm. With the increment of concentration, the proportion of DMTPA monomer gradually decreases while the large ones at tens of nanometers increases, demonstrating the progressively clustering process.

Fig. C3. DLS results of DMTPA/2-MTHF solutions with different concentrations.

Fig. C4. Emission spectra of 10^{-1} M DMTPA/2-MTHF solution with different λ_{ex} s.

Previously, we have identified the monomeric fluorescence (305 nm) and phosphorescence (420 nm) of DMTPA in 2-MTHF. Herein, we newly measured the emission spectra of 0.1 M DMTPA/2-MTHF solution with different excitation wavelengths (λ_{ex} s). With λ_{ex} s of 254, 312 and 365 nm, the emission maxima are 322, 374 and 452 nm, respectively, indicating the presence of diverse emissive species including monomers, excimers and clusters (Fig. C4).

Similar to the concentrated solutions, there might be different clusters with distinct through-space electronic interactions in crystals. Meanwhile, different emissive clusters can contribute to the same emission peak. From the excitation spectra of 595 and 650 nm, together with their emission properties, there should be more clusters present in crystals. However, whether there are more forms of clusters cannot be precisely determined only by the excitation spectra.

Thanks again for the reviewer's questions.

Response to the Comments and Suggestions of Reviewer #3

The authors have diligently addressed the reviewers' comments, including my own, thereby strengthening the manuscript. Consequently, I recommend its publication in Nature Communications.

Many thanks for the reviewer's comments and appreciation of our work.

We are now sending our revised manuscript to your editorial office *via* the online submission system. Thank you and the reviewers very much for your review and consideration. We appreciate your spending precious time to handle our manuscript. We look forward to hearing from you.

Yours sincerely,

W. Z. Yuan Ph.D.

Reviewer #1 (Remarks to the Author):

The revised manuscript has properly addressed the comments from the reviewers, and I recommend it for publication.

Reviewer #2 (Remarks to the Author):

The authors have addressed the review's comments carefully, and the manuscript has been improved. Thus, I recommend its publication at present.

Ms **No.:** NCOMMS-23-51350B

Ms **Title:** Photochromic luminescence of organic crystals arising from subtle molecular rearrangement

Ms **Authors:** Zihao Zhao, Yusong Cai, Qiang Zhang, Anze Li, Tianwen Zhu, Xiaohong Chen & Wang
Zhang Yuan

Response to the Comments and Suggestions of Reviewer #1

The revised manuscript has properly addressed the comments from the reviewers, and I recommend it for publication.

Many thanks for the reviewer's comments and appreciation of our work.

Response to the Comments and Suggestions of Reviewer #2

The authors have addressed the review's comments carefully, and the manuscript has been improved. Thus, I recommend its publication at present.

Many thanks for the reviewer's comments and appreciation of our work.

We are now sending our revised manuscript to your editorial office *via* the online submission system. Thank you and the reviewers very much for your review and consideration. We appreciate your spending precious time to handle our manuscript. We look forward to hearing from you.

Yours sincerely,

W. Z. Yuan Ph.D.